# RNA-dependent stabilization of SUV39H1 at constitutive heterochromatin

**Whitney L Johnson[1†], William T Yewdell[1†], Jason C Bell[1], Shannon M McNulty[2], Zachary Duda[3,4], Rachel J O'Neill[3,4], Beth A Sullivan[2], Aaron F Straight[1*]**

[1]Department of Biochemistry, Stanford University School of Medicine, Stanford, United States; [2]Department of Molecular Genetics and Microbiology, Duke University Medical Center, Durham, United States; [3]Department of Molecular and Cell Biology, University of Connecticut, Storrs, United States; [4]Institute for Systems Genomics, University of Connecticut, Storrs, United States

**Abstract** Heterochromatin formed by the SUV39 histone methyltransferases represses transcription from repetitive DNA sequences and ensures genomic stability. How SUV39 enzymes localize to their target genomic loci remains unclear. Here, we demonstrate that chromatin-associated RNA contributes to the stable association of SUV39H1 with constitutive heterochromatin in human cells. We find that RNA associated with mitotic chromosomes is concentrated at pericentric heterochromatin, and is encoded, in part, by repetitive α-satellite sequences, which are retained in cis at their transcription sites. Purified SUV39H1 directly binds nucleic acids through its chromodomain; and in cells, SUV39H1 associates with α-satellite RNA transcripts. Furthermore, nucleic acid binding mutants destabilize the association of SUV39H1 with chromatin in mitotic and interphase cells – effects that can be recapitulated by RNase treatment or RNA polymerase inhibition – and cause defects in heterochromatin function. Collectively, our findings uncover a previously unrealized function for chromatin-associated RNA in regulating constitutive heterochromatin in human cells.

*For correspondence: astraigh@stanford.edu

†These authors contributed equally to this work

**Competing interests:** The authors declare that no competing interests exist.

## Introduction

The histone methyltransferase Su(var)3–9 was first identified in forward genetic screens as a suppressor of position effect variegation in *Drosophila melanogaster* (*Tschiersch et al., 1994*). Previous studies identified important functions for the evolutionarily conserved SUV39 proteins in the silencing of heterochromatin, as well as in chromosome segregation and cell division (*Ekwall et al., 1996*; *Melcher et al., 2000*; *Peters et al., 2001*). This family of chromatin-modifying enzymes includes Clr4 in fission yeast (*Nakayama et al., 2001*), as well as SUV39H1 and SUV39H2 in humans (*Rea et al., 2000*). SUV39 proteins catalyze the di- and tri-methylation of lysine 9 of histone H3 (H3K9me2/3), and these histone modifications are bound by chromodomain-containing proteins, including the SUV39 enzymes themselves and the HP1 family of proteins (*Al-Sady et al., 2013*; *Bannister et al., 2001*; *Lachner et al., 2001*; *Müller et al., 2016*; *Wang et al., 2012*). HP1 protein binding to H3K9me2/3 chromatin is then thought to drive chromatin compaction and transcriptional repression through oligomerization (*Canzio et al., 2011*; *Fan et al., 2004*; *Grewal and Jia, 2007*).

SUV39H1 and H3K9me3 are predominately associated with constitutive heterochromatin, which represses 'selfish' genetic elements and repetitive DNA to promote genomic stability (*Bulut-Karslioglu et al., 2014*; *Peters et al., 2001*). In many eukaryotes, constitutive heterochromatin is concentrated at the repetitive sequences flanking centromeres, and is termed pericentric heterochromatin. In fission yeast, disruption of pericentric heterochromatin causes chromosome cohesion defects and chromosome missegregation (*Bernard et al., 2001*); and in mammals, defective

**eLife digest** Each cell in a human body contains the same DNA sequence, which serves as a set of instructions for how the body should develop and operate. However, only certain sections of DNA are "active" at any particular time and in any given type of cell. When a section of DNA is active, cells make many copies of it using a molecule called RNA. When a section of DNA in inactive, very little RNA is made. Some sections of DNA must always be kept inactive to avoid damaging the cell.

DNA is packaged around proteins called histones, and enzymes that modify histones control which sections of DNA are switched on or off. One such modifying enzyme, called SUV39H1, is important for inactivating sections of DNA that could cause harm to the cell if they are active. Previous studies showed that the loss of SUV39H1 and related proteins cause abnormalities and cancer in mice. However, it is not clear how this enzyme identifies and inactivates the DNA it needs to target.

Johnson, Yewdell et al. studied SUV39H1 in human cells. The experiments show that RNA binds to the SUV39H1 enzyme and controls how it interacts with DNA. Specifically, Johnson, Yewdell et al. found that sections of DNA that are inactive can still make a small amount of RNA, and that this RNA tethers SUV39H1 to the DNA to keep the DNA switched off. Mutant forms of SUV39H1 that are unable to interact with RNA fall off the DNA, which allows DNA sequences that are normally switched off to become active.

The findings of Johnson, Yewdell et al. reveal a new role for RNAs in regulating whether DNA is switched on or off. The next step is to determine whether other enzymes that can also modify histones use the same mechanism to activate or inactivate DNA. Differences in how the activity of DNA is regulated between individuals plays a crucial role in generating the diversity we see in nature. Therefore, this work helps us to understand our basic biology and may provide new opportunities for treating disease.

pericentric heterochromatin and aberrant transcription of pericentric repeats are associated with genomic instability and cancer (*Peters et al., 2001*; *Ting et al., 2011*; *Zhu et al., 2011*). These defects in constitutive heterochromatin are most evident in SUV39H1 and SUV39H2 double knock-out mice, which exhibit reduced embryonic viability, small stature, chromosome instability, an increased risk of tumor formation, and male infertility owing to defective spermatogenesis (*Peters et al., 2001*). Human SUV39H1 has been implicated in a variety of complex biological processes such as DNA damage repair (*Alagoz et al., 2015*; *Ayrapetov et al., 2014*; *Zheng et al., 2014*), telomere maintenance (*García-Cao et al., 2004*; *Porro et al., 2014*), cell differentiation (*Allan et al., 2012*; *Scarola et al., 2015*), and aging (*Zhang et al., 2015*).

Despite the fundamental role of SUV39H1 and SUV39H2 in heterochromatin formation, it is largely unclear how these enzymes are localized at specific genomic sites to generate heterochromatin. Other chromatin modifiers – in addition to binding DNA, post-translationally modified histones, and other chromatin-associated proteins – depend on interactions with noncoding RNAs for their proper localization (*Margueron and Reinberg, 2011*; *Rinn and Chang, 2012*). In fission yeast, the localization of pericentric heterochromatin proteins, including the SUV39 homolog Clr4, relies on the RNAi machinery (*Bühler and Moazed, 2007*; *Grewal and Jia, 2007*; *Moazed, 2011*), and RNAi has also been implicated in heterochromatin formation in other eukaryotic systems as well (*Fukagawa et al., 2004*; *Pal-Bhadra et al., 2004*). Recent studies reported that RNA is involved in targeting SUV39H1 to telomeres and to the *Oct4* locus (*Porro et al., 2014*; *Scarola et al., 2015*); however, it is unclear whether RNA plays a broader role in SUV39H1-dependent heterochromatin formation, and if direct RNA binding regulates the association of SUV39H1 with pericentric heterochromatin.

In this study, we establish that chromatin-associated RNA contributes to the localization of SUV39H1 at constitutive heterochromatin in humans. We find that RNA associates with the pericentric heterochromatin of human mitotic chromosomes in primary and immortalized cell lines, and that a portion of this RNA is encoded by pericentric α-satellite sequences. We show that SUV39H1 binds

without any observed sequence preference to both DNA and RNA in vitro, and that SUV39H1 binds RNA transcribed from pericentromeric repeats in human cells. Mutations that disrupt the nucleic acid binding function of SUV39H1 cause defects in its localization to pericentric heterochromatin, destabilize SUV39H1's association with chromatin, and result in heterochromatin silencing defects. We propose a model in which the direct binding of SUV39H1 to RNA and to methylated histones ensures proper constitutive heterochromatin function in humans.

## Results

### RNA associates with the pericentric regions of human mitotic chromosomes

Chromatin-associated RNA has a well-studied role in the formation of pericentric heterochromatin in fission yeast (*Bühler and Moazed, 2007*; *Grewal and Jia, 2007*; *Moazed, 2011*), but the role of RNA at human pericentric heterochromatin remains largely unexplored. To test if RNA is associated with pericentric heterochromatin in human cells, we used fluorescent pulse labeling of RNA to observe its localization on mitotic chromosomes. Because transcription is largely repressed in mitosis (*Gottesfeld and Forbes, 1997*), RNAs bound to mitotic chromosomes may be more likely to play regulatory roles than RNAs associated with chromatin during interphase. The morphology of condensed mitotic chromosomes also provides landmarks such as the primary centromere constriction and the telomere to facilitate the localization of RNAs.

We visualized chromosome-associated RNA by treating HeLa cells with the modified nucleoside ethynyl uridine (EU) (*Jao and Salic, 2008*), centrifuging mitotic chromosomes onto coverslips, and fluorescently labeling the RNA by coupling an azido-modified fluorophore to the alkyne group using copper-catalyzed cycloaddition (click chemistry). The EU-RNA signal we detected on human mitotic chromosomes, although distributed in distinct puncta along chromosome arms, was particularly concentrated around centromeres (*Figure 1A*). The RNA signal was sensitive to treatment with Ribonuclease (RNase) A, confirming that our EU treatment specifically labeled RNA; but not sensitive to RNase III or RNase H, indicating that this RNA possesses single-stranded (ssRNA) regions but not nuclease-accessible double-stranded RNA (dsRNA) or RNA-DNA hybrids (*Figure 1—figure supplement 1A*). Four out of eight different cell lines showed RNA enrichment at pericentric regions (*Figure 1—figure supplement 1B, C and D*). Both primary fibroblasts and HeLa cells exhibited pericentric RNA localization, indicating that this phenomenon is not specific to immortalized cells (*Figure 1—figure supplement 1B*).

### Characterization of pericentric RNA

A previous study demonstrated that transcription by RNA polymerase II (Pol II) can occur from the core centromere regions of human mitotic chromosomes (*Chan et al., 2012*). This RNA is presumably transcribed directly from the underlying centromeric DNA sequences, which in humans are composed of α-satellite repeats. α-satellite sequences, as well as other repeat classes – such as β-satellite and Satellite III – are also present at human pericentric regions (*Choo et al., 1992*; *Greig and Willard, 1992*). To determine which RNA sequences are localized at pericentric heterochromatin, we performed RNA FISH with different repeat-specific probes. We used DLD-1 cells for this analysis, as they have a more stable karyotype than HeLa cells and allow for more reliable chromosome identification. We saw no RNA FISH signal with β-satellite (chromosomes 13, 14, 15, 21, and 22) or Satellite III (chromosomes 14 and 22) probes despite observing clear DNA FISH signal (*Figure 1—figure supplement 1E*). However, we detected RNA FISH signal with two different α-satellite probes, one of which was specific for the pericentric α-satellite array D1Z5 (*Figure 1B and C*, *Figure 1—figure supplement 1F*) (*Pironon et al., 2010*). This signal was sensitive to RNase treatment (*Figure 1D*) and colocalized with H3K9me3 staining (*Figure 1C*, *Figure 1—figure supplement 1G*), but not with HEC1 staining that labels the core centromere/kinetochore (*Figure 1C*). We also observed that α-satellite RNA was localized in cis (i.e. at the same site of its transcription), as we saw that the RNA FISH signal was constrained to only the chromosomes known to contain the specific α-satellite DNA sequences recognized by our probes (*Figure 1D*, *Figure 1—figure supplement 1F and G*). Interestingly, even though we detected EU-RNA signal enriched at pericentric regions on HeLa chromosomes and not on DLD-1 chromosomes (*Figure 1A*, *Figure 1—figure supplement 1B*

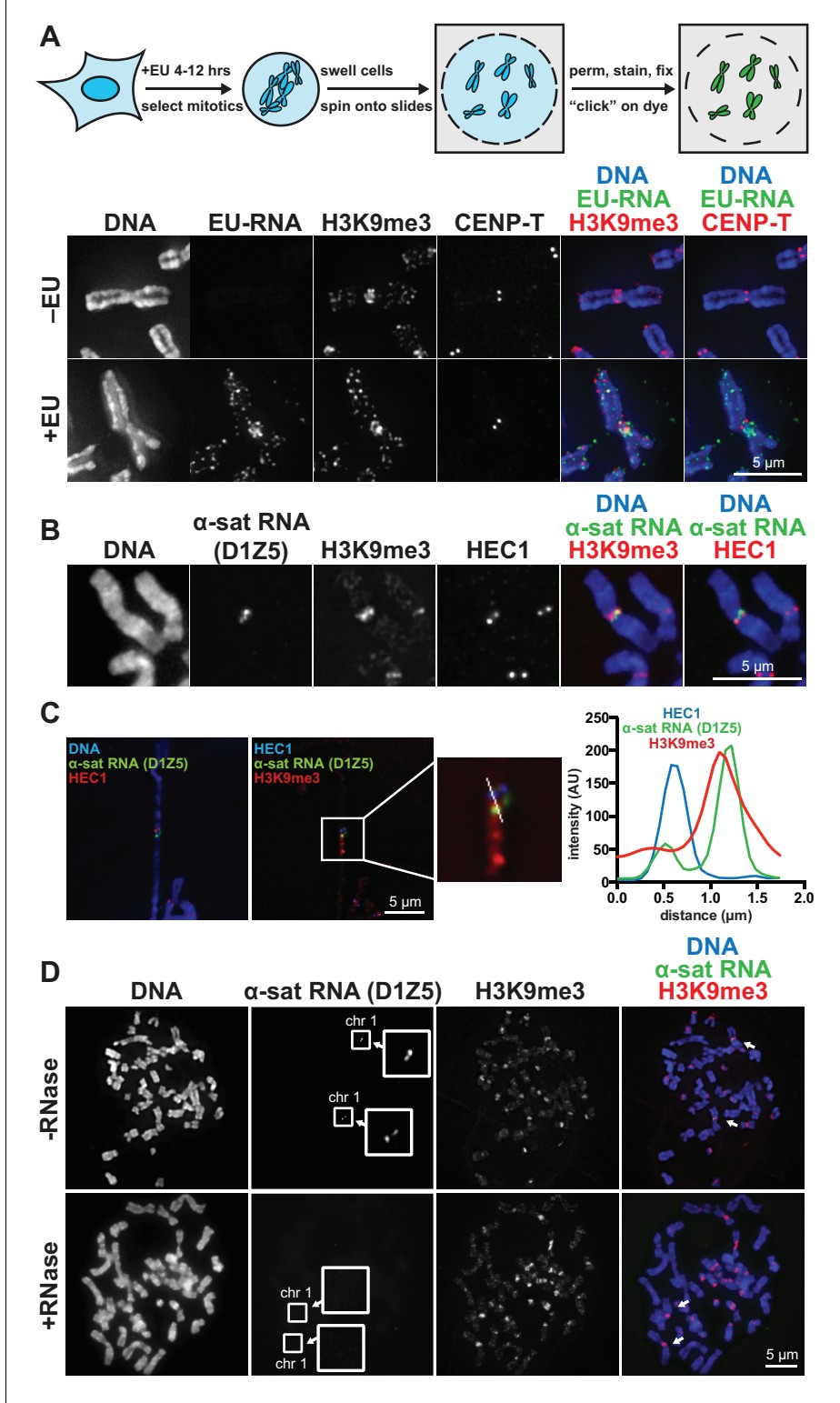

**Figure 1.** RNA associates with the pericentric regions of human mitotic chromosomes. (**A**) RNA localization on human mitotic chromosomes. A schematic of the chromosome-associated RNA labeling approach is shown at top. Mitotic HeLa cells were spun onto coverslips and stained for DNA (blue), ethynyl uridine labeled RNA (EU-RNA, green), and H3K9me3 (red) or CENP-T (red) to mark centromeres. Top row of images: cells were not treated with EU (-EU). Bottom row of images: cells were labeled with EU for 12 hr (+EU). (**B**) α-satellite RNA localization on mitotic chromosomes. Mitotic DLD-1 cells were spun onto coverslips, then α-satellite RNA was detected with a

*Figure 1 continued on next page*

*Figure 1 continued*

probe recognizing the pericentric D1Z5 array on human chromosome 1 (green). Chromosomes were also stained for DNA (blue), H3K9me3 (red), and HEC1 (red) antibodies to mark pericentric heterochromatin and the core centromere/kinetochore region. (C) D1Z5 α-satellite RNA overlaps with pericentric heterochromatin, but not the core centromere/kinetochore. α-satellite RNA FISH with the D1Z5 probe (green) on a stretched DLD-1 mitotic chromosome, co-stained for DNA (blue), H3K9me3 (red), and HEC1 (red). Line scans show α-satellite RNA overlaps with H3K9me3, but not HEC1. (D) RNase sensitivity of D1Z5 α-satellite RNA FISH signal. Spread mitotic DLD-1 cells were treated ± RNases, then stained for DNA (blue), D1Z5 α-satellite RNA (green), and H3K9me3 (red). See also *Figure 1—figure supplement 1 and 2*.

The following figure supplements are available for figure 1:

**Figure supplement 1.** Characterization of chromosome-associated pericentric RNA.

**Figure supplement 2.** Identifying the transcriptional requirements for chromosome-associated RNA.

and D), we detected α-satellite RNA by RNA FISH in both cell types (*Figure 1C and D*, *Figure 1—figure supplement 1F*), indicating that α-satellite RNA is localized at pericentric regions even in cells lines with low pericentric EU-RNA signal.

To determine which RNA polymerase transcribes the pericentric RNA, we added different polymerase inhibitors to cells during EU treatment. We measured a reduction of approximately 50% in pericentric RNA signal after treatment with α-amanitin or triptolide – which preferentially inhibit Pol II – but observed a complete loss of signal after treatment with actinomycin D or CX-5461 – which preferentially inhibit Pol I, and show no detectable Pol II inhibition (*Figure 1—figure supplement 2A and B*). Thus, most of the pericentric RNA signal we observe appears to be transcribed by Pol I. Total α-satellite RNA levels were not decreased by any RNA polymerase inhibition, suggesting that the amount of α-satellite transcribed in our 6 hr treatment window is likely a small percentage of the total α-satellite RNA (*Figure 1—figure supplement 2B*).

Our data does not exclude the possibility that some Pol II-dependent transcription is also occurring at the core centromere, as has been previously reported in human cells (*Chan et al., 2012*). In that study, incorporation of fluorescent nucleotides was only assayed on mitotic chromosomes, and because we label cells with EU for a longer time period (4–12 hr), we are likely observing a broader set of RNAs that are being transcribed before and during mitosis.

To investigate the cell cycle timing of pericentric RNA synthesis, we either labeled cells continuously prior to mitosis, or for a short pulse followed by washout prior to mitosis. We found that continuous EU labeling for 8 or 4 hr before mitosis led to pericentric EU-RNA signal, but a 4 hr EU pulse followed by a 4 hr chase resulted in no EU-RNA signal on mitotic chromosomes (*Figure 1—figure supplement 2C and D*). This suggests that the pericentric RNA we observe on mitotic chromosomes is transcribed within the 4 hr before mitosis, or that it is transcribed earlier but dissociates from chromosomes in the 4 hr washout period after labeling.

## SUV39H1 localization depends on chromatin-associated RNA

The finding that RNA is bound to the pericentric regions of human mitotic chromosomes is intriguing in light of previous observations that direct RNA binding by the heterochromatin factors Polycomb Repressive Complex 2 (PRC2) and HP1α is important for their proper localization and function (*Muchardt et al., 2002*; *Zhao et al., 2008*). Because SUV39 enzymes are the primary H3K9 methyltransferases acting at pericentric regions, we tested if the association of SUV39H1 with pericentric heterochromatin also depends on RNA. We digested chromosomes with RNase A and assessed SUV39H1 localization. In a stable HeLa cell line expressing GFP-tagged SUV39H1 under doxycycline inducible control, we induced expression with a 6 hr pulse of doxycycline to avoid SUV39H1 mislocalization to chromosome arms caused by overexpression (*Melcher et al., 2000*). We found that RNase A treatment reduced the pericentric localization of SUV39H1 to 43 ± 4% of the untreated control, and that SUV39H1 localization could be completely rescued by adding specific RNase inhibitors (*Figure 2A and B*). DNA staining at pericentric regions was not reduced in this experiment, indicating that the general chromosome structure at pericentric heterochromatin was not disrupted by

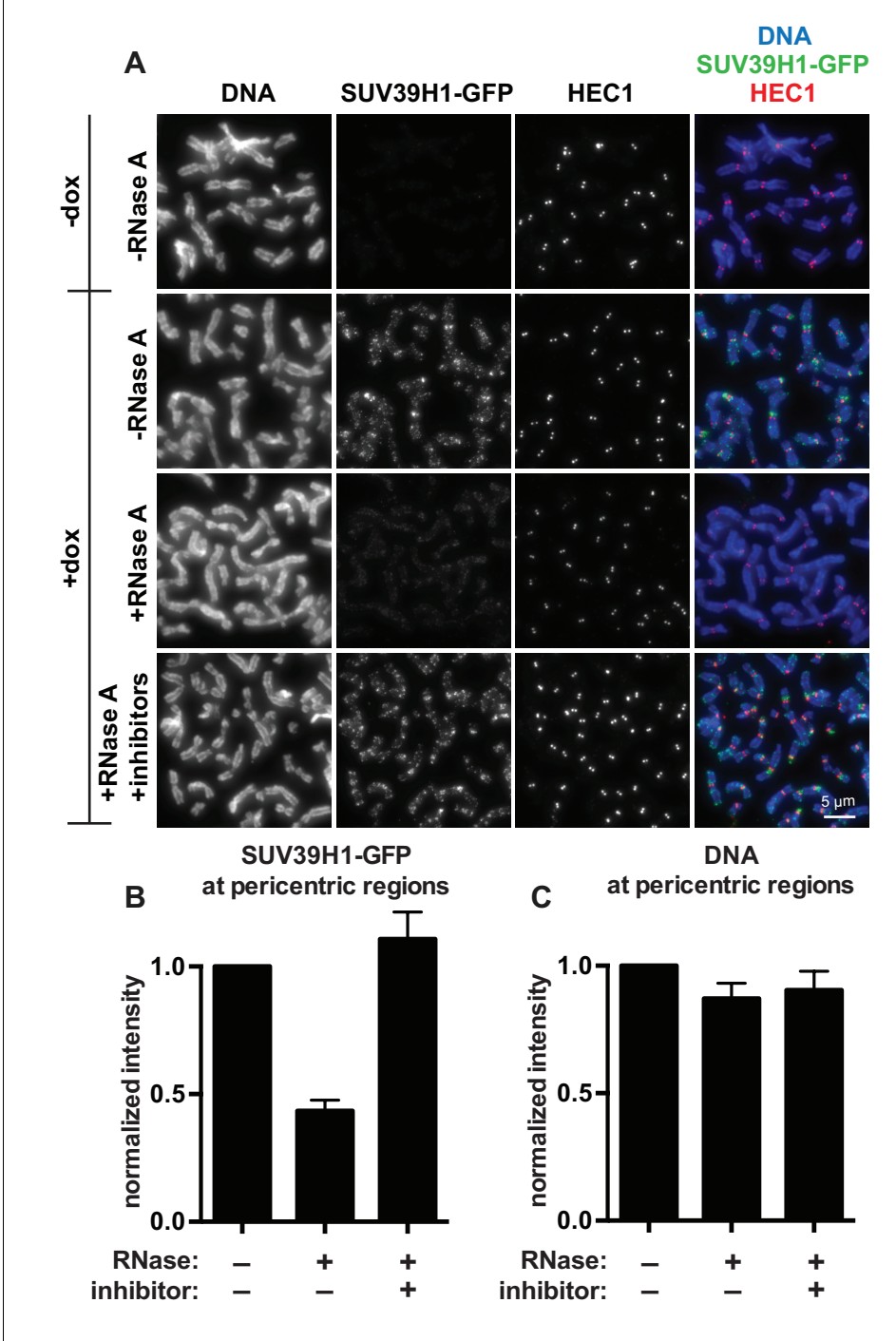

**Figure 2.** SUV39H1 localization on mitotic chromosomes is disrupted by RNase treatment. (**A**) Mitotic HeLa cells expressing or not expressing SUV39H1-GFP (+/-dox) were spread onto coverslips and incubated without RNase, with RNase A, or with RNase A plus RNase inhibitors. Cells were then stained for DNA (blue), anti-GFP to detect SUV39H1 (green), and HEC1 to mark centromeres (red). (**B**) Quantification of SUV39H1-GFP pericentric signal. The graph shows the average signal after subtracting background (-dox) and normalizing to control levels (+dox, -RNase A). n = 5 separate experiments, 15 cells quantified per condition per experiment, error bars are standard error. (**C**) Quantification of DNA pericentric signal. The graph shows the average signal after normalizing to control levels (+dox, -RNase A). n = 5 separate experiments, 15 cells quantified per condition per experiment, error bars are standard error. See also *Figure 2—figure supplement 1*.

*Figure 2 continued on next page*

*Figure 2 continued*

The following figure supplement is available for figure 2:

**Figure supplement 1.** Characterization of SUV39 DKO human cells.

RNase A treatment (*Figure 2C*). We also determined that total SUV39H1 protein levels are not affected by RNase A treatment (*Figure 2—figure supplement 1M*). These data demonstrate that SUV39H1 depends on the presence of RNA for its localization on human mitotic chromosomes.

SUV39H1 binds directly to HP1 proteins (*Yamamoto and Sonoda, 2003*), and mouse HP1α protein can directly bind RNA (*Muchardt et al., 2002*). Given that SUV39H1, HP1α, and RNA interact with one another, we tested the effect of SUV39 protein loss on HP1α and RNA localization. Using CRISPR/Cas9-mediated gene targeting, we generated SUV39H1 and SUV39H2 double knockout (SUV39 DKO) human HeLa and DLD-1 cell lines (*Figure 2—figure supplement 1*) that lack expression of both SUV39H1 and the partially redundant SUV39H2 proteins (*Figure 2—figure supplement 1A, B, H and I*). As observed in SUV39 double null mouse cells (*Lachner et al., 2001*; *Peters et al., 2001*), both H3K9me3 levels and HP1α localization were substantially reduced in human SUV39 DKO cells (*Figure 2—figure supplement 1A, D, E, H and K*). However, the levels of centromeric satellite RNA, including α-satellite RNA, increased in SUV39 DKO cells (*Figure 2—figure supplement 1C, F and J*) (*Lee et al., 1997*), consistent with SUV39H1/SUV39H2 loss causing defective heterochromatin and transcriptional depression of centromeric satellites. In contrast to the derepression of satellite RNA, we saw no significant changes in global RNA expression levels (*Figure 2—figure supplement 1G*). In the absence of SUV39H1 and SUV39H2, α-satellite RNA continued to localize to pericentric heterochromatin as assayed by RNA FISH (*Figure 2—figure supplement 1L*). Together, our data is consistent with a model in which RNA transcribed from pericentric α-satellite sequences maintains an association with the site of transcription during mitosis. In the absence of the SUV39 enzymes, α-satellite transcripts remain localized at pericentric regions, but transcriptional silencing, HP1α localization, and H3K9me3-dependent heterochromatin are disrupted.

## SUV39H1 directly binds nucleic acids through its chromodomain

To determine whether SUV39H1 can bind RNA directly, we performed electrophoretic mobility shift assays (EMSAs) with a random 19-mer ssRNA oligonucleotide and MBP-tagged human SUV39H1 protein purified from *E. coli*. Because we were unable to purify full-length SUV39H1 protein free of degradation products, we purified truncations of SUV39H1 containing either the N-terminal extension alone (amino acids 1–41), the chromodomain alone (amino acids 42–106), the N-terminal extension plus the chromodomain (amino acids 1–106), or the C-terminus consisting of the pre-SET, SET, and post-SET domains (amino acids 107–412) (*Figure 3A and B*). We found that the SUV39H1 truncations containing the chromodomain bound to RNA with the highest affinity, whereas the C-terminal fragment showed no detectable binding (*Figure 3C and D*). The addition of the N-terminal extension to the chromodomain of SUV39H1 (1–106) provided a 16-fold increase in affinity relative to the chromodomain alone (42-106), with dissociation constants ($K_d$) of 0.15 ± 0.01 μM and 2.31 ± 0.17 μM, respectively. The N-terminal extension alone bound to RNA with a significantly reduced $K_d$ (51 ± 30 μM) compared to the truncations containing the chromodomain. Excess unlabeled RNA competed with radiolabeled RNA for binding to SUV39H1 (*Figure 3—figure supplement 1A*), confirming that the RNA gel shift was not due to irreversible aggregation. From these data, we conclude that the chromodomain, SUV39H1 42–106, is sufficient for RNA binding activity. This is consistent with previous findings that chromodomains can act as RNA binding domains (*Akhtar et al., 2000*), and that the telomeric TERRA RNA associates with SUV39H1 (*Porro et al., 2014*).

## SUV39H1 can bind multiple nucleic acid types with minimal sequence preference

To determine if purified SUV39H1 exhibited a binding preference for different types of nucleic acids, we tested its binding to ssRNA, ssDNA, dsRNA, dsDNA and RNA/DNA hybrids using 50 nucleotides

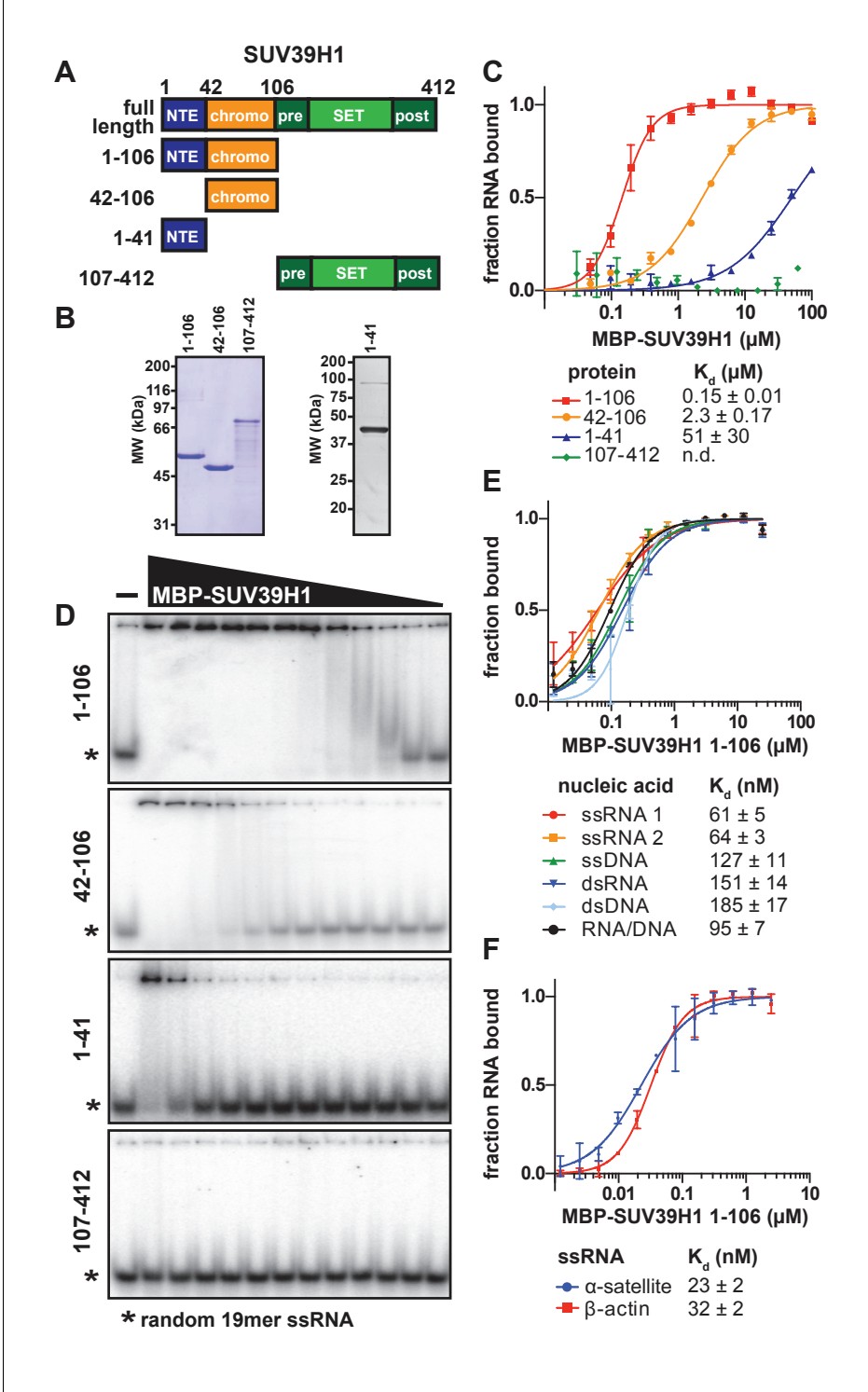

**Figure 3.** SUV39H1 directly binds nucleic acids through its chromodomain. (**A**) Domain schematic of SUV39H1 truncations. NTE, N-terminal extension; chromo, chromodomain; pre, pre-SET; post, post-SET. Amino acid residues are listed above and to the left of each truncation. (**B**) Coomassie-stained gel of purified human SUV39H1 truncations fused to MBP. (**C**) Quantification of SUV39H1 domains binding to 19mer RNA. Binding curves are from quantifying EMSAs shown in 3D. Error bars are standard deviation from two independent experiments. Dissociation constants ($K_d$, $\mu$M) displayed on graph are determined by non-linear fitting of the binding curves. Standard error represents the error of the curve fitting to the average of the two experimental replicates. (**D**) Representative EMSAs showing binding of purified MBP-SUV39H1 truncations with a 19mer RNA oligo (*), 1–41,

*Figure 3 continued on next page*

*Figure 3 continued*

42–106 and 1–106 diluted 2-fold from 100 μM, 107–412 diluted 2-fold from 62 μM. Quantified in **3C**. (**E**) Binding of SUV39H1 to all nucleic acid types. Binding curves are from quantifying EMSAs (***Figure 3—figure supplement 1C***) of MBP-tagged SUV39H1 1–106 binding to 50mer nucleic acids (ssRNA, ssDNA, dsRNA, dsDNA, or RNA/DNA) (***Figure 3—figure supplement 1B***). Various nucleic acids are composed of the first 50 bases of *E. coli* maltose binding protein (MBP): ssRNA1, sense MBP 1–50; ssRNA 2, anti-sense MBP 1–50; ssDNA, sense MBP 1–50. Error bars, dissociation constants ($K_d$, μM), and standard error calculated as in **3C**. (**F**) Quantification of MBP-SUV39H1 1–106 binding to 180mer α-satellite or β-actin ssRNA (representative EMSA in ***Figure 3—figure supplement 1E***). Error bars, dissociation constants ($K_d$, μM), and standard error calculated as in 3C. See also ***Figure 3—figure supplement 1***.

The following figure supplement is available for figure 3:

**Figure supplement 1.** Characterization of SUV39H1-RNA binding in vitro.

---

of the *E. coli* MBP sequence to generate each nucleic acid type (***Figure 3—figure supplement 1B***). We found that SUV39H1 1–106 bound to all five types of nucleic acids with similar affinities, ranging from 61 ± 5 nM for ssRNA to 185 ± 17 nM for dsDNA (***Figure 3E***, ***Figure 3—figure supplement 1C***). SUV39H1 1–106 bound to the sense and antisense strand of MBP 1–50 ssRNA with similar affinities (61 ± 5 nM ssRNA1 vs. 64 ± 3 nM ssRNA2) (***Figure 3E***), and showed a 2.5-fold difference in affinity for the sense and anti-sense 19mer ssRNA used in ***Figure 3C and D*** (145 ± 9 nM sense vs. 366 ± 22 nM anti-sense) (***Figure 3—figure supplement 1D***).

The pericentric regions of human chromosomes are composed of repetitive α-satellite DNA sequences, which are weakly transcribed in humans (***Ideue et al., 2014***; ***Wong et al., 2007***). Because SUV39H1 localizes to pericentric regions (***Aagaard et al., 1999***) and we detected chromosome-associated α-satellite RNA at these sites (***Figure 1C and D***, ***Figure 1—figure supplement 1F***), we tested whether SUV39H1 preferentially binds to α-satellite RNA. We measured SUV39H1 binding to a ssRNA containing a single monomeric unit of α-satellite (180 nucleotides), compared to a segment of β-actin mRNA of equal length. We found that SUV39H1 1–106 bound α-satellite or β-actin RNA with similar affinities: 23 ± 2 nM and 32 ± 2 nM, respectively (***Figure 3F***, ***Figure 3—figure supplement 1E***). This is consistent with the sequence-independent binding we observed for 19mer and 50mer ssRNA oligonucleotides. We conclude that SUV39H1 can bind both double- and single-stranded nucleic acids, has minimal sequence preference for the sequences tested thus far, and that binding affinity increases with increasing nucleic acid length (***Figure 3—figure supplement 1F***).

## Identification of SUV39H1 RNA binding deficient mutants

Our observation that RNA is bound to pericentric heterochromatin at the same sites where SUV39H1-dependent histone methylation occurs suggests that direct RNA binding may regulate the localization and function of SUV39H1. Recent studies have proposed that direct RNA binding by SUV39H1 is necessary for its localization to specific genomic locations (***Porro et al., 2014***; ***Scarola et al., 2015***), but this model has not been tested directly. To test whether RNA binding by SUV39H1 controls its interaction with chromatin, we identified RNA binding deficient mutants of SUV39H1 by alanine-scanning mutagenesis of the SUV39H1 chromodomain (42-106) (***Figure 4—figure supplement 1A and B***). We identified mutants that either decreased (12 mutants) or increased (11 mutants) the affinity of SUV39H1 for ssRNA more than 10-fold when compared to the wild-type (WT) SUV39H1 chromodomain (***Figure 4—figure supplement 1B***). Interestingly, 6 of the 12 mutants that decreased RNA binding affinity removed a positively charged residue (R or K), and 7 of the 11 mutants that increased RNA binding removed a negatively charged residue (D or E) (***Figure 4—figure supplement 1B***). By performing a more comprehensive analysis of the binding of these 23 mutants (***Figure 4A***, ***Figure 4—figure supplement 1C***), we discovered two mutations, R55A and R84A, that almost completely abolished RNA binding by SUV39H1, reducing the affinity over 40-fold compared to WT (***Figure 4C***, ***Figure 4—figure supplement 1C and D***).

Because SUV39H1 can bind all nucleic acids with comparable affinities, it is likely that the R55A and R84A mutations that disrupt RNA binding also prevent binding to DNA. Therefore, we refer to these mutations as nucleic acid binding mutations.

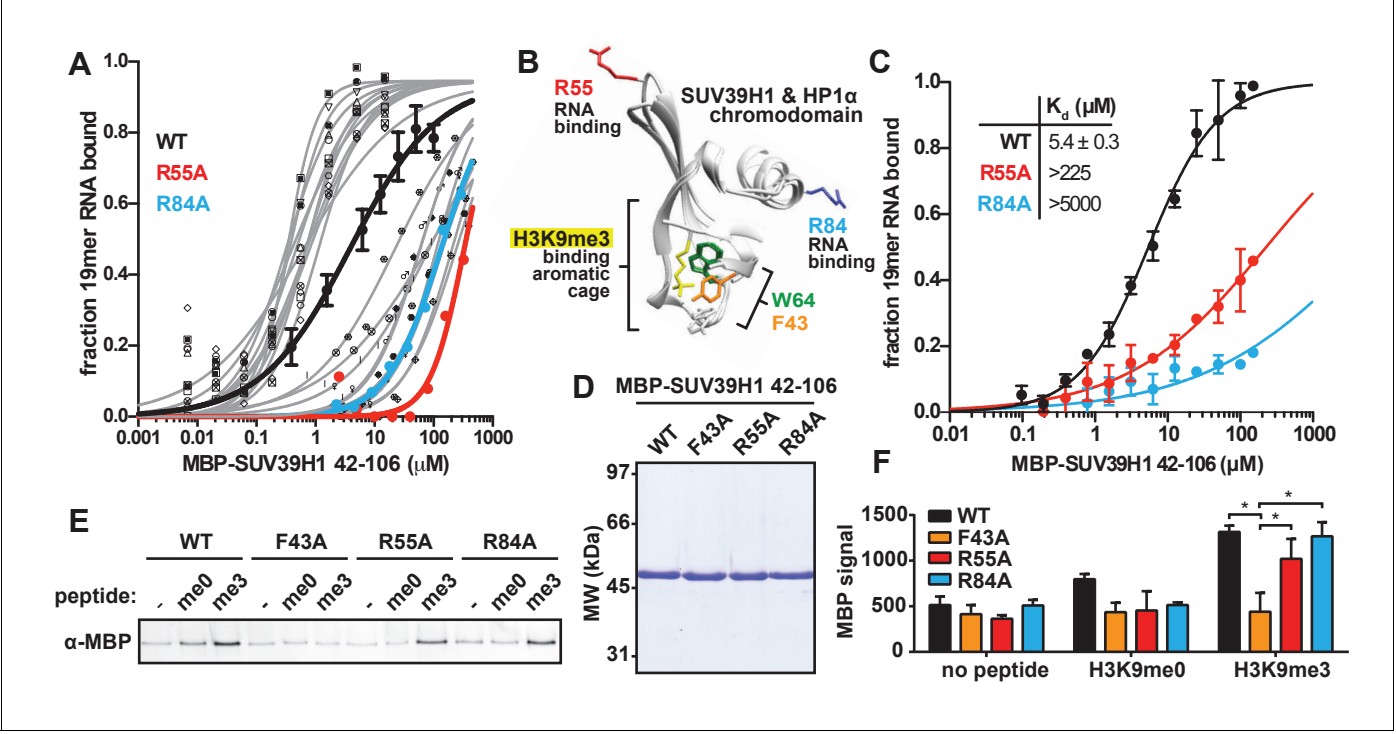

**Figure 4.** Identification and biochemical characterization of SUV39H1 RNA binding-deficient mutants. (A) Identification of SUV39H1 mutants that affect RNA binding. Binding curves from second round of EMSA screening showing the binding of purified MBP-SUV39H1 42–106 point mutants to 19mer RNA (table of measured dissociation constants in *Figure 4—figure supplement 1B*). (B) Crystal structures of the chromodomains of human SUV39H1 (aa 44–91) (*Wang et al., 2012*) and human HP1α (aa 18–68) bound to H3K9me3 peptide (yellow) (*Kaustov et al., 2011*). Residues in SUV39H1 important for RNA binding (red and blue) or H3K9me3 binding (green and orange) are highlighted. The H3K9me3 peptide was co-crystalized with HP1α, but not SUV39H1. (C) Quantification of WT SUV39H1, SUV39H1[R55A], and SUV39H1[R84A] binding to 19mer RNA. Binding curves generated by quantifying filter binding assays shown in *Figure 4—figure supplement 1D*. Error bars are standard deviation from three independent experiments. Dissociation constants ($K_d$, µM) displayed on graph are determined by non-linear fitting of the binding curves. Standard error represents the error of the curve fitting to the average of the three experimental replicates. (D) Coomassie stained gel of purified MBP-SUV39H1 42–106 proteins – WT, F43A, R55A, or R84A. (E) Binding of SUV39H1 mutants to H3K9me3. Representative α-MBP western blot showing the amount of purified MBP-SUV39H1 42–106 protein, WT or indicated mutant, bound to streptavidin beads conjugated to either H3K9me0 (me0), H3K9me3 (me3), or no peptide (-) added. (F) Quantification of western blot shown in 4E, error bars are standard deviation, n = 3, *p<0.03. See also *Figure 4—figure supplement 1*.

The following figure supplement is available for figure 4:

**Figure supplement 1.** In vitro characterization of SUV39H1 chromodomain mutants.

## Mutational separation of nucleic acid binding from methylated histone recognition

The chromodomains of HP1 proteins specifically recognize methylated H3K9 through three conserved aromatic residues, called the 'aromatic cage' (*Bannister et al., 2001*; *Jacobs and Khorasanizadeh, 2002*; *Lachner et al., 2001*; *Nielsen et al., 2002*). The SUV39H1 chromodomain also specifically binds to methylated H3K9 peptides, most likely via three similar aromatic cage residues (*Figure 4B*) (*Wang et al., 2012*). The R55A and R84A mutations that disrupt nucleic acid binding are distant from the aromatic cage in the 3D structure of SUV39H1, and thus we hypothesized that they do not affect H3K9me3 binding (*Figure 4B*) (*Wang et al., 2012*). We tested this by measuring the binding of WT and mutant SUV39H1 chromodomains to unmethylated (me0) or trimethylated (me3) H3K9 peptides (*Figure 4D,E and F*). As previously reported, we found that the WT SUV39H1 chromodomain preferentially binds to H3K9me3 compared to H3K9me0, and this binding is completely disrupted by the aromatic cage mutant SUV39H1[F43A] (*Figure 4E and F*) (*Wang et al., 2012*). Importantly, neither of the nucleic acid binding mutants (SUV39H1[R55A] and SUV39H1[R84A]) alter the binding

of SUV39H1 to H3K9me3 (*Figure 4E and F*). We conclude that SUV39H1[R55A] and SUV39H1[R84A] mutants specifically disrupt the interaction between SUV39H1 and nucleic acids without perturbing binding to H3K9me3.

We also confirmed that the R55A, R84A, and F43A mutations had no effect on the methyltransferase activity of full-length SUV39H1 (*Figure 4—figure supplement 1E*). Interestingly, the addition of RNA inhibits SUV39H1 activity, as has been previously shown for PRC2 and several other SET domain histone methyltransferases (*Cifuentes-Rojas et al., 2014*; *Kaneko et al., 2014*). However, SUV39H1 nucleic acid binding mutants were equally inhibited by RNA, suggesting that this inhibition is largely independent of direct RNA binding.

## SUV39H1 associates with α-satellite RNA in human cells

Our observations that RNA is associated with pericentric heterochromatin and that SUV39H1 directly binds RNA in vitro prompted us to test if SUV39H1 associates with RNA in human cells. In SUV39H1-GFP expressing HeLa cells, SUV39H1 colocalizes with EU-labeled RNA at pericentric regions of mitotic chromosomes (*Figure 5A*). To test if SUV39H1 can bind to α-satellite RNA, we expressed GFP-tagged WT SUV39H1, the nucleic acid binding deficient mutant SUV39H1[R55A], or GFP alone under doxycycline inducible control. We then crosslinked the cells with formaldehyde, immunoprecipitated the GFP-tagged protein, and analyzed SUV39H1-associated RNAs by reverse transcription followed by quantitative PCR (RT-qPCR). We found that α-satellite RNA was enriched in the WT SUV39H1 IP 13 ± 2 fold when compared to GFP alone (*Figure 5B and C*, *Figure 5—figure*

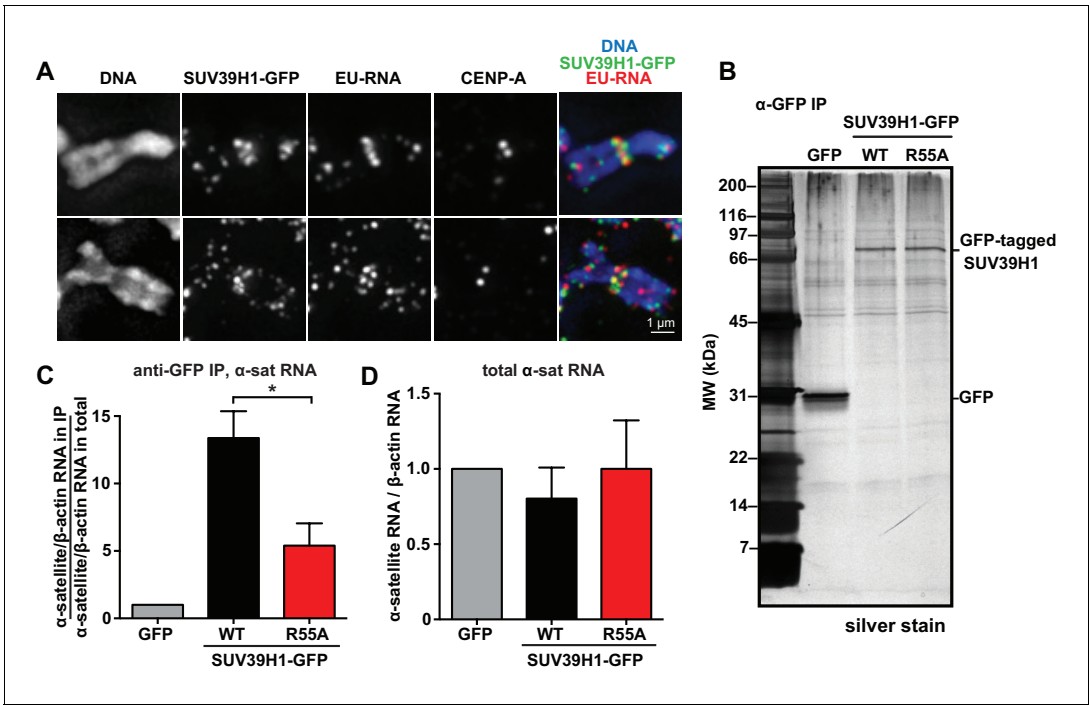

**Figure 5.** SUV39H1 association with α-satellite RNA in vivo depends on direct nucleic acid binding. (**A**) Colocalization of SUV39H1 and RNA at pericentric regions. SUV39H1-GFP expressing HeLa cells were labeled with EU and induced with doxycycline for 6 hr, then mitotic cells were spun onto coverslips. Cells were stained for DNA (blue), with anti-GFP to detect SUV39H1-GFP (green), for EU-RNA (red), and CENP-A to mark centromeres. (**B**) Silver stained gel showing protein immunoprecipitated by an anti-GFP antibody from cell lines expressing GFP, SUV39H1-GFP, or SUV39H1[R55A]-GFP. (**C**) Quantification of α-satellite RNA immunoprecipitated with SUV39H1. RNA was isolated from IPs, and RT-qPCR was performed to detect α-satellite RNA sequence. Enrichment values are the ratio of α-satellite/β-actin RNA in the IP over the ratio of α-satellite/β-actin RNA in total lysate, normalized to the GFP values. Error bars are standard error, n = 3, *p<0.04. (**D**) Total α-satellite RNA levels in GFP, SUV39H1-GFP, and SUV39H1[R55A]-GFP cell lines, divided by the amount of β-actin RNA, normalized to the GFP values. See also *Figure 5—figure supplement 1*.

The following figure supplement is available for figure 5:

**Figure supplement 1.** Immunoprecipitation of SUV39H1-GFP from human cells.

*supplement 1A*) and that R55A mutation reduced the amount of α-satellite RNA to 45 ± 16% of the levels bound to WT (*Figure 5C*). The expression of WT SUV39H1 or SUV39H1$^{R55A}$ did not lead to changes in total α-satellite RNA levels in the presence of endogenous SUV39H1, indicating that the difference in immunoprecipitated α-satellite RNA was not due to changes in the overall levels of α-satellite RNA (*Figure 5D*). In addition, the R55A mutation did not disrupt SUV39H1's association with HP1α, indicating that α-satellite RNA binding to SUV39H1 is not occurring indirectly through HP1α (*Figure 5—figure supplement 1B*). These results demonstrate that SUV39H1 associates with α-satellite RNA in cells, and that this association is facilitated by the nucleic acid binding activity of the SUV39H1 chromodomain.

## SUV39H1 relies on direct RNA binding for its localization to pericentric heterochromatin

To test how the SUV39H1 chromodomain interactions with RNA and H3K9me3 contribute to its localization in human cells, we measured the extent of localization of GFP-tagged WT or mutant SUV39H1 on spread mitotic chromosomes. The R55A and R84A nucleic acid binding mutants of SUV39H1 showed defects in localization, reducing SUV39H1 signal by about half (R55A: 42 ± 8% of WT levels, R84A: 45 ± 9% of WT levels) (*Figure 6A, B and C*); indicating that the direct interaction of SUV39H1 with nucleic acids is important for its localization to pericentric heterochromatin. We also found that the aromatic cage mutants W64A and F43A led to more pronounced localization defects than those caused by the nucleic acid binding mutants (W64A: 10 ± 4% of WT levels, F43A: 15 ± 5% of WT levels) (*Figure 6A, B and C*). These results suggest that both H3K9me3 binding and nucleic acid binding contribute to SUV39H1 localization to pericentric heterochromatin during mitosis.

To specifically test the contribution of RNA to SUV39H1 localization, we digested mitotic chromosomes with RNase A, a treatment we previously observed removed both pericentric RNA (*Figure 1D*, *Figure 1—figure supplement 1A*) and WT SUV39H1 (*Figure 2A and B*) from chromosomes. RNase A treatment reduced the pericentric localization of SUV39H1 to 43 ± 4% of the undigested control, similar to the levels of the untreated SUV39H1$^{R55A}$ mutant in the same experiments (52 ± 9% of WT) (*Figure 6D and E*). This indicates that the effect of the R55A mutation on SUV39H1 localization can be fully attributed to a loss in RNA binding, independent of DNA binding. This is further supported by the fact that RNase A treatment did not significantly reduce the amount of SUV39H1$^{R55A}$ at pericentric chromatin (untreated: 52 ± 9%, RNase A: 42 ± 5%) (*Figure 6D and E*). Taken together, these results demonstrate that direct binding to both chromosome-associated RNA and H3K9me2/3 are required for proper localization of SUV39H1 to pericentric heterochromatin.

## SUV39H1 can recruit HP1α to pericentric heterochromatin independently of H3K9 methylation

Previous studies found that HP1α localization at heterochromatin is sensitive to RNase treatment in both mouse interphase cells and on human mitotic chromosomes (*Maison et al., 2002*; *Wong et al., 2007*). We also observed a reduction in HP1α levels on chromatin after RNase A treatment, to 77 ± 13% of control levels (*Figure 6—figure supplement 1A and B*). We wondered if the RNase sensitivity of HP1α could be due in part to its interaction with SUV39H1. Consistent with this, expressing SUV39H1-GFP for 6 hr, which causes an increase in SUV39H1-GFP at pericentric regions, also caused a 1.5-fold increase in HP1α localization (*Figure 6—figure supplement 1A and B*), in the absence of any detectable change in H3K9me3 levels (*Figure 6—figure supplement 1C*). Additionally, we observed that RNase A treatment abolished the increase in HP1α localization following exogenous SUV39H1-GFP expression (*Figure 6—figure supplement 1A and B*). Finally, after expressing SUV39H1 mutants, the extent of HP1α localization positively correlated with SUV39H1 localization (*Figure 6A and C*; *Figure 6—figure supplement 1D and E*). These results are consistent with a model in which the pericentric localization of HP1α during mitosis depends on the SUV39H1 protein – independent of H3K9 methylation – and that HP1α localization is sensitive to RNase treatment because SUV39H1 localization depends on direct binding to RNA.

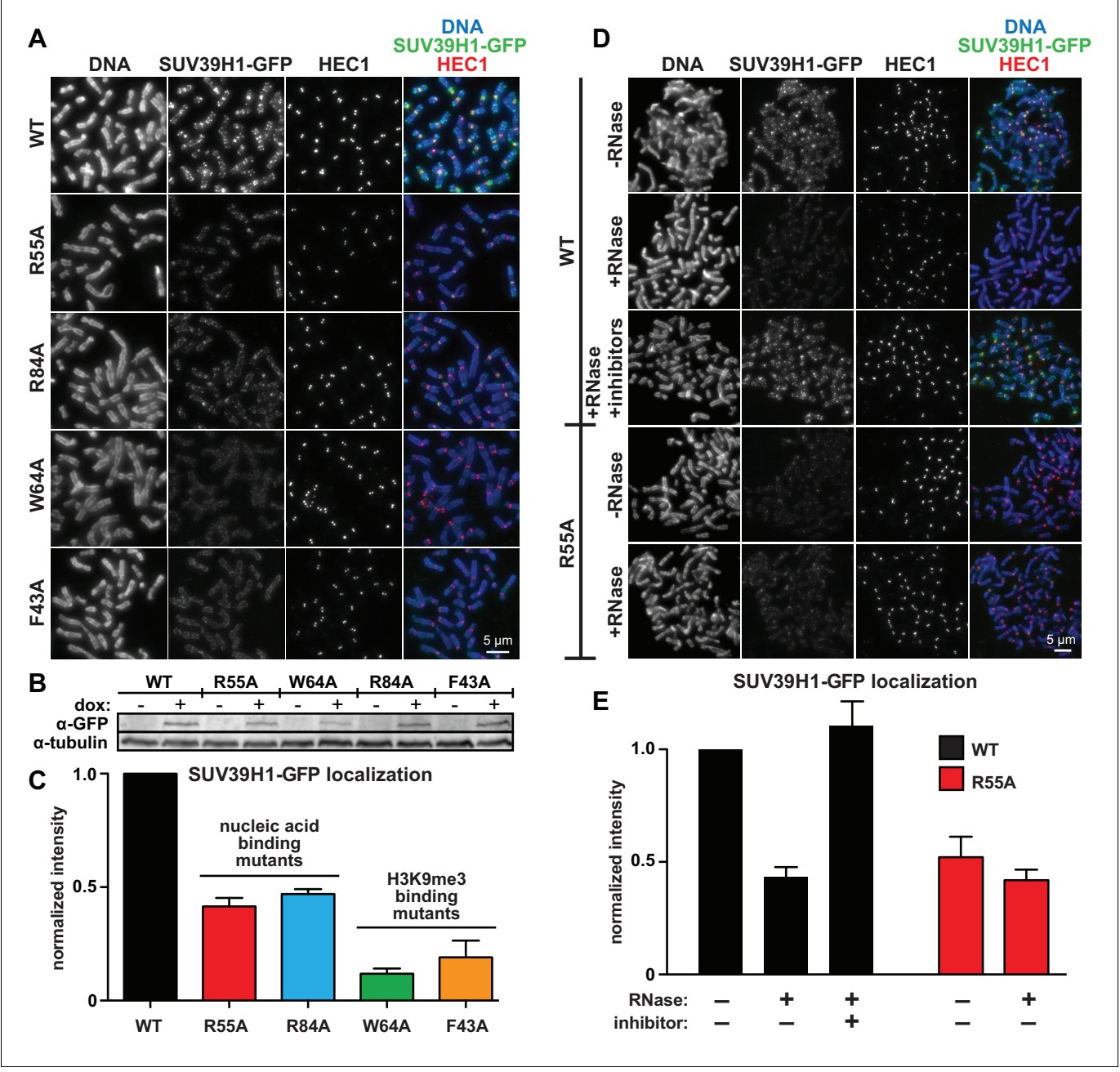

**Figure 6.** Direct nucleic acid binding by SUV39H1 regulates its localization on mitotic chromosomes. (**A**) Localization of WT or mutant SUV39H1-GFP on human mitotic chromosomes. Expression of SUV39H1-GFP was induced in HeLa cell lines for 6 hr, then mitotic cells were spun onto coverslips and stained for DNA (blue), with anti-GFP to detect SUV39H1-GFP (green), and for HEC1 to mark centromeres (red). Shown are representative images. (**B**) Western blot assessing expression of inducible SUV39H1-GFP in HeLa cell lines after 6 hr of +/- doxycycline induction. (**C**) Quantification of SUV39H1-GFP levels at pericentric regions, from experiment shown in 6A. Each experiment was normalized to the WT measurement. Bars show the average of n = 3 separate experiments, 15 cells quantified per condition per experiment, error bars represent standard error. (**D**) Localization of WT or mutant SUV39H1-GFP on human mitotic chromosomes after RNase treatment. Mitotic HeLa cells expressing SUV39H1-GFP (WT or R55A) were spread onto coverslips and incubated without RNase, with RNase A, or with RNase A plus RNase inhibitors. Cells were then stained for DNA (blue), anti-GFP to detect SUV39H1 (green), and HEC1 to mark centromeres (red). Shown are representative images. (**E**) Quantification of SUV39H1-GFP levels at pericentric regions after RNase treatment, from experiment shown in 6D. Each experiment was normalized to the untreated WT measurement. Shown are the averages of n = 5 separate experiments, 15 cells quantified per condition per experiment, error bars represent standard error. See also *Figure 6—figure supplement 1*.

The following figure supplement is available for figure 6:

*Figure 6 continued on next page*

*Figure 6 continued*

**Figure supplement 1.** HP1α localization correlates with SUV39H1 localization, despite no change in H3K9me3 levels.

## SUV39H1 depends on direct nucleic acid binding for its stable association with heterochromatin

The requirement for RNA for SUV39H1 localization in mitosis suggests that RNA binding might generally stabilize the interaction of SUV39H1 with chromatin. To assess this in interphase cells, we measured the turnover rate of GFP-tagged WT SUV39H1, SUV39H1$^{R55A}$, SUV39H1$^{R84A}$, SUV39H1$^{F43A}$, and SUV39H1$^{W64A}$ using fluorescence recovery after photobleaching (FRAP) (*Figure 7A, B and C*). After photobleaching, WT SUV39H1 recovered with an average half-time of $27 \pm 7$ s (*Figure 7D*), with a stable, immobile fraction of SUV39H1 that did not recover during the time frame of our experiment (*Figure 7B, C and E*). The mobile fraction represented $58 \pm 7\%$ of the total nuclear SUV39H1 (*Figure 7E*), consistent with previous measurements (*Hahn et al., 2013*; *Krouwels et al., 2005*). The aromatic cage mutants SUV39H1$^{W64A}$ (*Figure 7B*) and SUV39H1$^{F43A}$ (*Figure 7C*), which disrupt binding to H3K9me3, had a much faster recovery half-time of $6 \pm 2$ and $5 \pm 1$ s, respectively (*Figure 7D*). Unlike WT SUV39H1, approximately 90–95% of each aromatic cage mutant was determined to be mobile within the nucleus (*Figure 7E*). Together, this high level of mobility and rapid recovery is consistent with weak and/or transient associations with chromatin that are normally stabilized by SUV39H1's interaction with methylated histones.

Interestingly, the nucleic acid binding deficient mutants SUV39H1$^{R55A}$ (*Figure 7B*) and SUV39H1$^{R84A}$ (*Figure 7C*) showed an average recovery time in between that of WT SUV39H1 and the H3K9me3 binding mutants, with half-lives of $14 \pm 3$ and $13 \pm 2$ s, respectively (*Figure 7D*). Consistent with this trend, both SUV39H1$^{R55A}$ and SUV39H1$^{R84A}$ demonstrated a level of mobility ($77\% \pm 6$) that was greater than WT SUV39H1, but less than the aromatic cage mutants SUV39H1$^{W64A}$ and SUV39H1$^{F43A}$ (*Figure 7E*). We noted that the sub-nuclear localization of the aromatic cage mutants was more diffuse, and generally lacking the typical heterochromatin centers observed in the WT SUV39H1, SUV39H1$^{R55A}$ and SUV39H1$^{R84A}$ cell lines (*Figure 7A*). Combined with our in vitro biochemical analysis, these observations support a model in which direct binding to nucleic acids stabilizes the association of SUV39H1 with chromatin in interphase cells as well as in mitosis.

If RNA stabilizes SUV39H1 on chromatin, then inhibiting RNA transcription to reduce the levels of chromatin-associated RNA is expected to destabilize SUV39H1 localization. When we added the RNA Pol I inhibitor CX-5461 – a treatment that we saw reduces RNA localization on mitotic chromosomes (*Figure 1—figure supplement 2A and B*) – the fluorescence recovery half-time of WT SUV39H1 decreased from $27 \pm 7$ s to $16 \pm 3$ s, comparable to the half-time of the nucleic acid binding mutant SUV39H1$^{R55A}$ (*Figure 7F*). However, we also observed a slight effect of Pol I inhibition on SUV39H1$^{R55A}$, with a decrease in half-time from $14 \pm 3$ to $11 \pm 2$ s. This suggests that RNA polymerase inhibition may have non-specific effects on SUV39H1 localization in addition to disrupting SUV39H1 binding to chromatin through chromatin-associated RNAs. We also saw similar effects with α-amanitin or triptolide treatment (*Figure 7—figure supplement 1A*), suggesting that Pol II may also be playing a role in SUV39H1 localization; however, this effect may also be due to the partial inhibition of Pol I with these inhibitors (*Figure 1—figure supplement 2B*). Although polymerase inhibition resulted in measureable changes in the recovery kinetics of SUV39H1, we observed no significant changes in overall SUV39H1 mobility (*Figure 7G*).

## SUV39H1 interactions with both nucleic acids and H3K9me3 are necessary for proper heterochromatin silencing

Because RNA binding and H3K9me3 binding both contribute to SUV39H1 localization, we wanted to determine the effect of disrupting these interactions on SUV39H1-dependent heterochromatic silencing. Using CRISPR/Cas9 genome editing, we generate DLD-1 cell lines in which the endogenous SUV39H1 was altered to contain either a nucleic acid binding mutation (R55A), an H3K9me3 binding mutation (F43A) or both (R55A/F43A) (*Figure 8A*). We made these SUV39H1 mutations in a SUV39H2 knockout background to prevent redundant activities of SUV39H2 from masking effects of

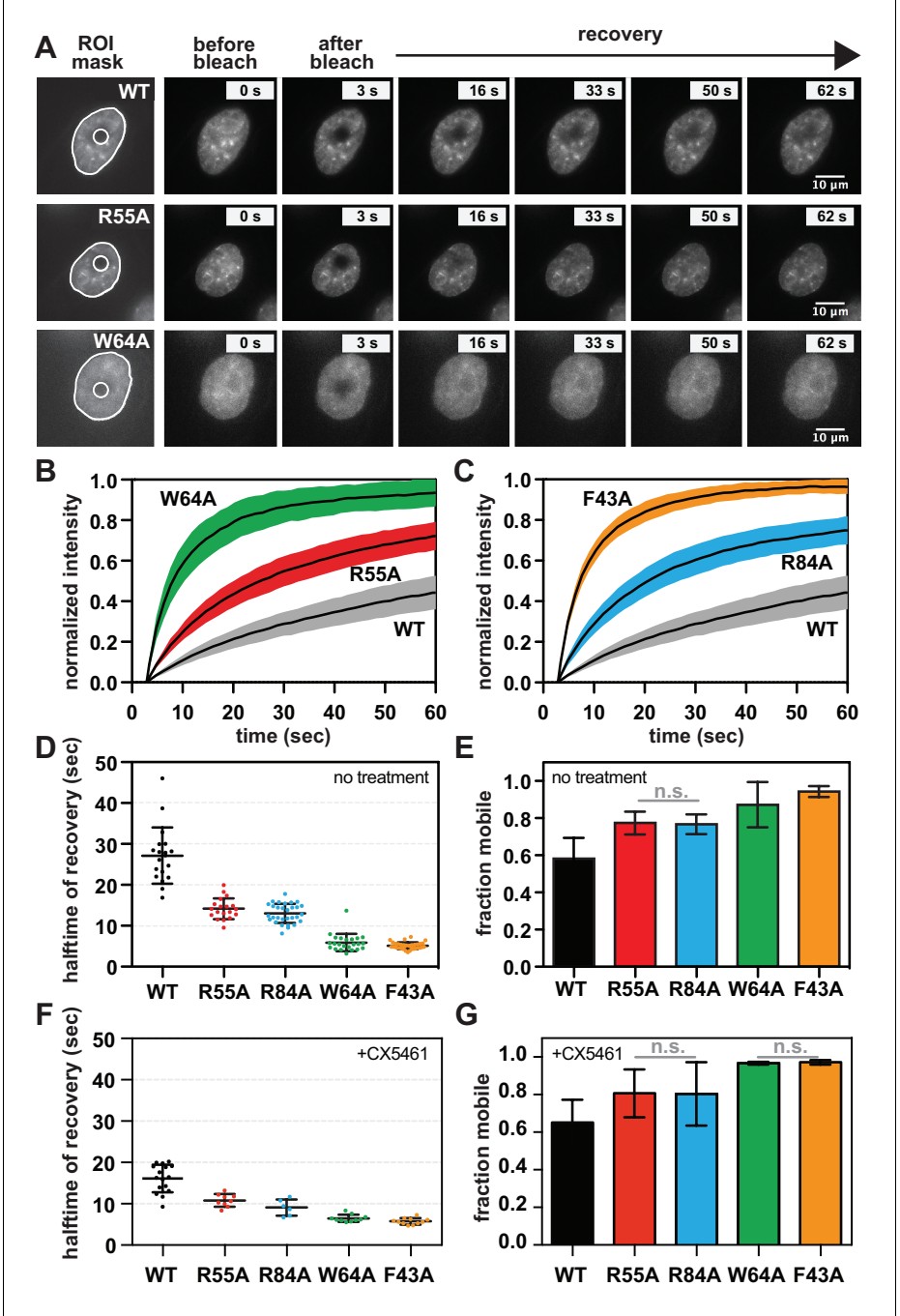

**Figure 7.** SUV39H1 depends on direct nucleic acid binding for its stable association with chromatin in interphase cells. (A) Fluorescence recovery after photobleaching (FRAP) assay to measure SUV39H1-GFP association with chromatin. Montage of images of representative cells expressing either WT or mutant SUV39H1-GFP, as indicated, in our FRAP assay. Leftmost panel shows the ROIs used for quantification and analysis, with the innermost circle representing the diameter of the photobleached area. (B, C) Quantification of SUV39H1-GFP recovery after photobleaching. The fluorescence intensity of the bleached area was normalized relative to the whole nucleus, and plotted as a function of time for WT (gray, n = 19), R55A (red, n = 19), W64A (green, n = 23), R84A (blue, n = 23) and F43A (orange, n = 35) mutants. Black lines are the average between individual traces and colored shading represents standard deviation. WT curves are shown in both panels for clarity. (D) Half-time to recovery measurements from SUV39H1-GFP FRAP traces. Individual traces (averages shown in 7B and 7C) were fit to a single exponential, and the half-time to recovery for each curve was plotted for WT and mutant SUV39H1-GFP cells. (E) Quantification of the mobile fraction of SUV39H1-GFP. Bar graph of the mean amplitude of the curves

*Figure 7 continued on next page*

*Figure 7 continued*

from 7B and 7C. Error bars represent standard deviation. The differences between mobile fractions are statistically significant (p<0.01), except between R55A and R84A. (**F**) Half-time to recovery measurements from SUV39H1-GFP FRAP traces, in the presence of the RNA Pol I inhibitor CX5461. Individual traces were fit to a single exponential, and the half-time to recovery for each curve was plotted for WT and mutant SUV39H1-GFP cells. (**G**) Quantification of the mobile fraction of SUV39H1-GFP, in the presence of the RNA Pol I inhibitor CX5461. Bar graph of the mean amplitude of recovery curves. Error bars represent standard deviation. See also *Figure 7—figure supplement 1*.

The following figure supplement is available for figure 7:

**Figure supplement 1.** Effect of transcription inhibition on SUV39H1 fluorescence recovery, and model for RNA-dependent retention of SUV39H1 at heterochromatin.

these mutations. We found that total H3K9me3 levels, as well as H3K9me3 localization at centromeres, were reduced in all SUV39H1 mutants compared to WT SUV39H1 (*Figure 8B, C, D and E*), demonstrating a role for both H3K9me3 binding and nucleic acid binding by SUV39H1 in maintaining normal H3K9 methylation levels. To assess the role of SUV39H1 nucleic acid binding and H3K9me3 binding in repressing pericentromeric heterochromatin, we measured α-satellite RNA levels in the WT and mutant SUV39H1 cell lines. We found that α-satellite expression increased 1.35-fold, 1.56-fold, and 2.10-fold, in the R55A, F43A, and the R55A/F43A double mutant cells lines, respectively, compared to WT (*Figure 8F*). Interestingly, although no change in H3K9me3 was detected between the F43A single mutant and the R55A/F43A double mutant (*Figure 8C and D*), a significant increase in α-satellite RNA levels was observed in R55A/F43A compared to F43A (*Figure 8F*), indicating that SUV39H1 nucleic binding may make H3K9me3-independent contributions to heterochromatin function. Importantly, because the defects in SUV39H1 localization caused by the R55A nucleic acid binding mutation were similar to defects caused by RNase treatment or RNA polymerase inhibition (*Figure 6E*, *Figure 7*), we support a model in which the heterochromatin defects we observe with this mutant are due to breaking interactions with chromatin-associated RNA. Taken together, these data demonstrate that both RNA binding and H3K9me3 binding by SUV39H1 contribute to silencing of constitutive heterochromatin in human cells.

## Discussion

Several models have been proposed for how chromatin-associated RNAs regulate the organization and transcription of the genome (*Cifuentes-Rojas et al., 2014*; *Davidovich et al., 2013*; *Kaneko et al., 2014*; *Keller et al., 2013*; *Rinn and Chang, 2012*). One emerging theme is that chromatin-associated RNAs help to localize chromatin-modifying proteins, either by recruiting them to target loci in trans (*Kotake et al., 2011*; *Zhao et al., 2008*; *Zheng et al., 2015*) or by stabilizing their interactions with chromatin in cis (*Maison et al., 2002*, *2011*; *Sigova et al., 2015*). Here, we demonstrate that α-satellite RNAs are attached to the pericentric regions of human mitotic chromosomes in cis. These RNAs regulate the localization of the histone methyltransferase SUV39H1 via a direct protein-RNA interaction, ensuring the stable association of SUV39H1 with pericentric heterochromatin.

### α-satellite RNA at human pericentric heterochromatin

We show for the first time that RNAs associated with mitotic chromosomes are enriched at pericentric regions in several human cell lines (*Figure 1*, *Figure 1—figure supplement 1*). Previous studies have shown that transcription by RNA polymerase II (Pol II) can occur at the centromeres of human mitotic chromosomes (*Chan et al., 2012*; *Liu et al., 2015*). The RNA we observe is distinct from this previously described centromeric RNA in that its localization is insensitive to Pol II inhibition (*Figure 1—figure supplement 2A and B*), and exhibits broader pericentric localization. RNA FISH shows that the RNA we observe is transcribed from pericentric α-satellite sequences outside of the core centromere region (*Figure 1D*, *Figure 1—figure supplement 1G*). Additionally, the α-satellite RNAs we detect are associated in cis with the chromosomes from which they are transcribed (*Figure 1D*, *Figure 1—figure supplement 1F and G*), supporting the idea that their localization may provide

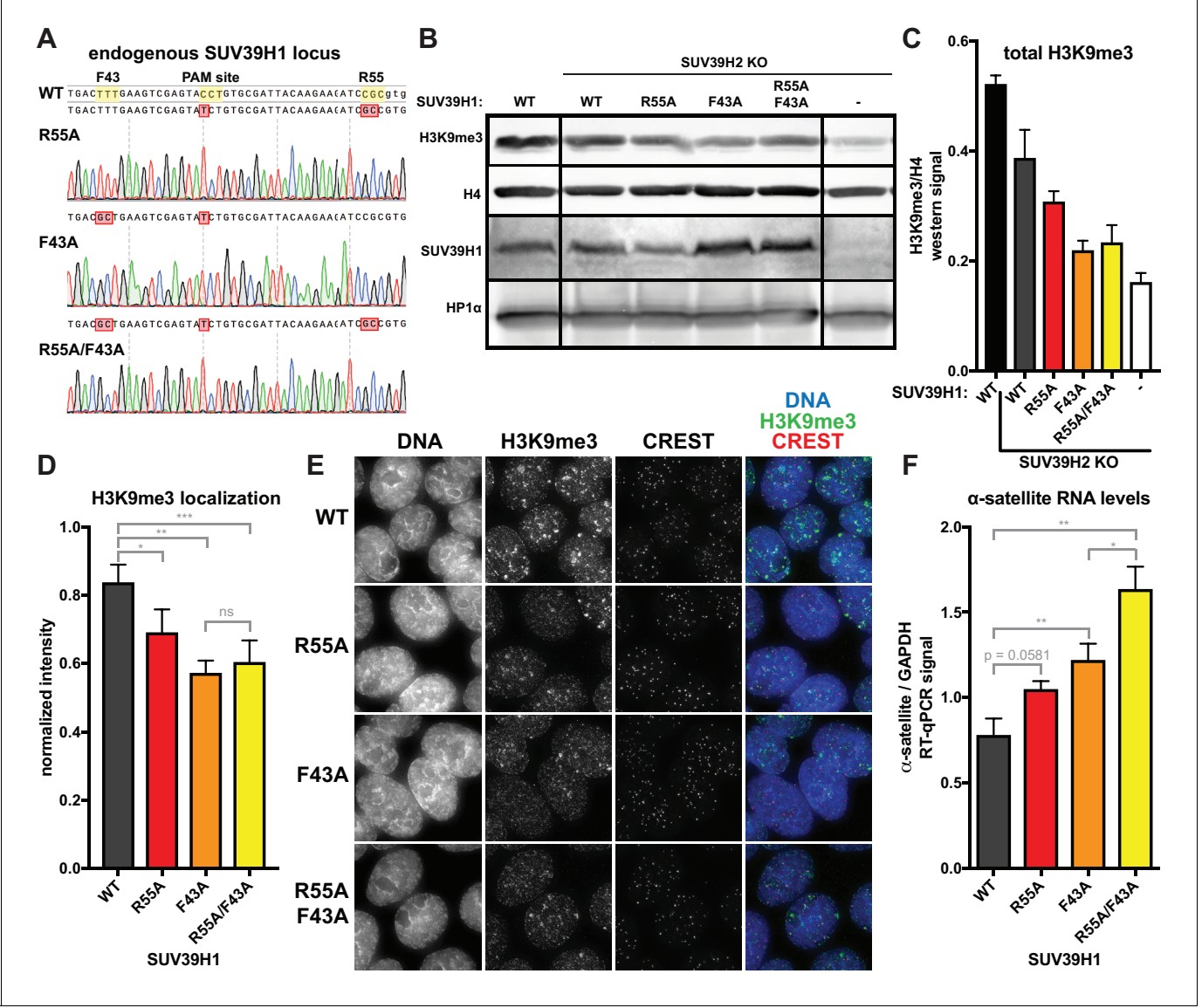

**Figure 8.** Nucleic acid binding and H3K9me3 binding both contribute to SUV39H1-mediated constitutive heterochromatin silencing. (**A**) Sanger DNA sequencing traces showing point mutations in the endogenous SUV39H1 locus in DLD-1 cells, generated by CRISPR/Cas9 gene editing. (**B**) Representative western blot showing H3K9me3, histone H4, SUV39H1, and HP1α levels in normal unedited DLD-1 cells (far left), or SUV39H2 KO DLD-1 cells with the indicated mutations in SUV39H1. (**C**) Total H3K9me3 levels in mutant SUV39 DLD-1 cells, measured by quantitative western blot. Nuclear lysate was prepared and blotted with indicated antibodies, and H3K9me3 signal was normalized to histone H4 levels. Graphed are the means of 3 repeats, errors bars represent standard error. (**D**) Quantification of H3K9me3 localization at centromeres in mutant SUV39 DLD-1 cells. Cells were grown on coverslips, fixed and permeabilized, then stained with the indicated antibodies, imaged, and quantified using centromere finder software, using CREST staining as a centromere marker. Graphed are the means of 4 repeats, error bars represent standard error. Significance was determined using paired, two-tailed t-tests. *p<0.05, **p<0.005, ***p<0.0005. (**E**) Representative images of immunofluorescence staining of mutant SUV39 DLD-1 cells, used for quantification shown in 8D. Cells are stained for DNA (blue), H3K9me3 (green), and CREST centromere stain (red). (**F**) Quantification of α-satellite RNA in mutant SUV39 DLD-1 cells by RT-qPCR. Total α-satellite RNA levels are normalized to GAPDH RNA levels. Graphed are the means of 3 repeats, error bars represent standard error. Significance was determined using paired, two-tailed t-tests. *p<0.05, **p<0.005.

direct feedback on the local transcriptional state. This differs from the phenomenon observed in *Drosophila* cells, where SAT III RNA transcribed from the X chromosome localizes to all centromeres in trans (*Rošić et al., 2014*).

It is interesting that we observe RNA on mitotic chromosomes, as transcription is silenced during mitosis (*Gottesfeld and Forbes, 1997*), and known noncoding RNAs, such as Xist, dissociate from

chromatin during mitosis in human cells (*Hall et al., 2014*). This may reflect the particular importance of centromeric and pericentric RNA in maintaining chromatin structure around the centromere for chromosome cohesion and proper chromosome segregation during mitosis (*Ekwall et al., 1996*; *Melcher et al., 2000*; *Peters et al., 2001*). Centromeric and pericentric RNAs have been proposed to play functional roles in several different organisms (*Blower, 2016*; *Carone et al., 2009*; *Choi et al., 2011*; *Grenfell et al., 2016*; *Probst et al., 2010*; *Rošić et al., 2014*), and the transcription of repetitive α-satellite sequences at human centromeres has been implicated in maintaining centromere identity (*Chan et al., 2012*; *Quénet and Dalal, 2014*; *Wong et al., 2007*), regulating the mitotic kinase Aurora B (*Ferri et al., 2009*; *Ideue et al., 2014*; *Jambhekar et al., 2014*; *Mallm and Rippe, 2015*), and influencing chromosome stability (*Zhu et al., 2011*) and cohesion (*Liu et al., 2015*). RNAs encoded by pericentric DNA may maintain their association with mitotic chromosomes to regulate these processes, and may interact with chromatin in a manner that is distinct from other regulatory RNAs that dissociate from chromatin during mitosis.

## Characterization of the SUV39H1-RNA interaction

We find that the human SUV39H1 protein directly binds to RNA through its chromodomain. This is consistent with the identification of other chromodomain-containing proteins that bind RNA (*Akhtar et al., 2000*) and the interaction of the SUV39H1 chromodomain with TERRA RNA (*Porro et al., 2014*). However, we also detect binding of SUV39H1 to other types of nucleic acid species (dsDNA, ssDNA, dsRNA, RNA:DNA) (*Figure 3E*), observe no strong preference for any sequence tested (*Figure 3E and F*, *Figure 3—figure supplement 1D*), and see a positive correlation between nucleic acid length and binding affinity (*Figure 3—figure supplement 1F*). Additionally, we find that a loss of positively charged residues generally decreases SUV39H1-RNA affinity, whereas a loss of negatively charged residues generally increases SUV39H1-RNA affinity (*Figure 4—figure supplement 1B and C*). One hypothesis that accounts for these observations is that binding occurs via electrostatic interactions between the negatively charged phosphate backbone of nucleic acids and the positively charged residues within the SUV39H1 chromodomain.

It is notable that the Polycomb Repressive Complex 2 (PRC2) also exhibits relatively promiscuous, length-dependent, sequence-independent RNA binding; though specificity for certain RNA sequences can be observed in vivo and in vitro (*Davidovich et al., 2013*, *2015*). It is possible that for SUV39H1 and PRC2, and potentially for other chromatin modifiers, RNA binding may generally stabilize interactions with chromatin rather than target the proteins to specific sites. Although we observe no sequence specificity for binding α-satellite RNA in vitro (*Figure 3F*), SUV39H1 associates with α-satellite RNA in human cells, and this interaction is impaired by mutations that disrupt the ability of SUV39H1 to directly bind RNA in vitro (*Figure 5C*). It will be important to further explore the specificity of the SUV39H1-RNA interaction, as well as more broadly identify RNAs that associate with SUV39H1 in human cells, as had been done with PRC2 (*Davidovich et al., 2013*; *Kaneko et al., 2013*).

## Role of nucleic acid binding in SUV39H1 localization and function

By generating point mutations that separate the nucleic acid binding and H3K9me3 binding functions of the SUV39H1 chromodomain, we assessed the individual contributions of these two interactions on SUV39H1 localization and function. We show for the first time that direct nucleic acid binding contributes to SUV39H1 localization by stabilizing its interaction with heterochromatin, in both interphase and mitotic human cells (*Figures 6* and *7*), and that the localization defects we observe in RNA binding mutants of SUV39H1 are largely recapitulated by RNase treatment or RNA polymerase inhibition (*Figures 6D, E*, *7F and G*); strongly suggesting that SUV39H1 requires direct RNA binding for its stable association with chromatin. Importantly, we demonstrate that breaking the nucleic acid binding of SUV39H1 leads to defects in HP1α localization (*Figure 6—figure supplement 1*), H3K9me3 methylation (*Figure 8B, C, D and E*), and repression of repetitive α-satellite sequences (*Figure 8F*).

It is important to note that because of the high conservation between SUV39H1 and HP1α aromatic cages (*Figure 4B*), it has been widely hypothesized that an intact aromatic cage is necessary for the localization of SUV39H1 (*Wang et al., 2012*), but this hypothesis has not been directly assessed. Both aromatic cage mutations of SUV39H1 we tested resulted in an almost complete loss

of SUV39H1 localization and retention on chromatin, confirming that binding to H3K9me3 through the aromatic cage is indeed a key determinant of SUV39H1 localization (*Figures 6* and *7*).

## A model for the RNA-dependent stabilization of SUV39H1 on chromatin

It remains a major outstanding question how SUV39H1 specifically recognizes its genomic sites of action. Although it had been originally proposed that HP1 proteins directly recruit SUV39H1 (*Bannister et al., 2001*; *Hall et al., 2002*; *Lachner et al., 2001*), subsequent studies in fission yeast demonstrated that SUV39/Clr4 localization is largely independent of HP1 proteins (*Motamedi et al., 2008*; *Sadaie et al., 2004*). A recent study also showed that the N-terminus of mammalian SUV39H1 contributes to its association with chromatin in vitro and in vivo through a zinc-finger like motif thought to directly bind DNA (*Müller et al., 2016*). However, SUV39H1 binds a variety of repetitive sequences in mammals (*Bulut-Karslioglu et al., 2014*) with no apparent DNA consensus sequence, making it unlikely that DNA recognition is the sole factor determining SUV39H1 targeting. In addition, the chromodomain of SUV39H1 has been shown previously to be important for its localization (*Melcher et al., 2000*), but the relative contributions of binding to methylated H3K9 versus other interactions mediated by the chromodomain were not assessed.

Here, we identify both RNA and H3K9me2/3 as main determinants of SUV39H1 localization. Importantly, the RNA-dependent mechanism for SUV39 function appears to be present in both human and mouse cells (*Shirai et al., 2017*; *Velazquez Camacho et al., 2017*) suggesting a conserved role for RNA in regulating constitutive heterochromatin. A possible advantage to binding chromatin-associated RNA is that it provides direct feedback on the transcriptional state of the locus, ensuring that over-transcribed repeats retain more SUV39H1 to promote repression. Because SUV39H1 displays no apparent sequence preference for binding RNA (*Figure 3E and F*, *Figure 3—figure supplement 1D*), it is unlikely that RNA binding alone could confer selective binding for one genomic region over another. It is also unlikely that H3K9me3 binding alone targets SUV39H1, for many H3K9me3 sites exist in the mammalian genome, but not all recruit SUV39H1 (*Bulut-Karslioglu et al., 2014*).

We propose a model in which SUV39H1 engages chromatin through several different interactions that in combination define its genomic distribution (*Figure 7—figure supplement 1C*). When direct binding to RNA is disrupted, SUV39H1 can still bind to H3K9me3; however, SUV39H1's interaction with chromatin is destabilized, and its localization at heterochromatin is reduced. RNA binding may help SUV39H1 distinguish between target and non-target sites in the genome that both have H3K9me2/3. In addition to directly binding RNA (*Figure 3*) (*Porro et al., 2014*), H3K9me3 (*Figure 4E and F*) (*Wang et al., 2012*), and DNA (*Figure 3E*) (*Müller et al., 2016*), SUV39H1 also directly binds the deacetylase Sirtuin 1 (SIRT1) (*Bosch-Presegué et al., 2011*; *Vaquero et al., 2007*), and associates with the retinoblastoma tumor suppressor (Rb) protein (*Nielsen et al., 2001*; *Vandel et al., 2001*). These and other interactions may influence SUV39H1 recruitment, stable association, and spread on chromatin – as well as its methyltransferase activity – to define its sites of action. Thus, a major question going forward is how the collective interactions between SUV39H1 and its binding partners give rise to locus specific targeting and heterochromatin formation. Towards this goal, we have demonstrated that H3K9me2/3 and chromatin-associated RNA are key determinants for SUV39H1 localization and are necessary for maintaining silenced constitutive heterochromatin in human cells.

## Materials and methods

### Cell culture

HeLa, U2OS, HFF, HT1080, Huh-7, and primary fibroblasts were grown in Dulbecco's Modified Eagle Medium (DMEM), hTERT-RPE cells were grown in DMEM/F12 media, and DLD-1 cells were grown in RPMI 1640 media. HeLa cells and U2OS cells were acquired from the ATCC. We obtained DLD-1 human colorectal cancer cells from Dr. Daniele Fachinetti and Dr. Don Cleveland, primary fibroblasts from Dr. Paul Khavari, HFF (human foreskin fibroblast) cells from Dr. Matthew Bogyo, HT-1080 fibrosarcoma cells from Dr. Jianghong Rao, hTERT RPE-1 (retina pigmented epithelial) cells from Dr. Tim Stearns, Huh-7 human hepatoma cells from Dr. Peter Sarnow, and Flp-In T-REx HeLa cells from Dr.

Pat Brown. These cells were not independently authenticated. Mycoplasma contamination was monitored frequently by cytoplasmic DAPI staining. All cells were grown under American Tissue Culture Collection (ATCC) standard conditions. All media was purchased from Thermo Fisher and supplemented with 10% vol/vol FBS and 100 U/ml penicillin and streptomycin.

## Generation of engineered human cell lines

Flp-In T-REx HeLa cell lines expressing SUV39H1-GFP proteins were made by co-transfecting pOG44 Flp-recombinase expression vector and pcDNA5/FRT/TO tet inducible Flp-In cloning vector (Invitrogen) containing SUV39H1-GFP sequences (ASP 1751, 2137, 2668, 2663, 2670, 2789) using Fugene 6 (Promega). After 48–72 hr, cells were selected with 350 µg/mL hygromycin B (Invitrogen) and 15 µg/mL blasticidin S (Invivogen) until visible colonies formed, then the entire selected population of cells was saved.

SUV39 DKO cells were generated essentially as described (*Ran et al., 2013*). Briefly, HeLa or DLD-1 cells were transfected with pX458 (pSpCas9(BB)−2A-GFP) containing SUV39H1 and SUV39H2 gRNA sequences using Fugene HD (Promega). SUV39H1 gRNA sequence (ASON 2868, ASP 2985): 5'-caccgGTTCCTCTTAGAGATACCGA-3'. SUV39H2 gRNA sequence (ASON 2878, ASP 2990): 5'-caccgAAAGCTCTACAAGATGGCGG-3'. After 4–5 days, GFP-positive cells were single-cell sorted into 96-well plates using a Sony SH800 cell sorter. Clonal populations were expanded and screened by western blotting for SUV39H1. Mutations in SUV39H1 and SUV39H2 were confirmed by purifying genomic DNA, PCR amplifying the targeted locus, and analyzing the amplicons by high throughput sequencing on a MiSeq system (Illumina).

DLD-1 cell lines with point mutations in the endogenous SUV39H1 locus were generated by first making a SUV39H2 KO cell line (ASTC 269, described above), then performing further genome editing by co-transfecting pX458 (pSpCas9(BB)−2A-GFP) with a SUV39H1-targeting gRNA (ASP 2987; 5'-caccgGGATCTTCTTGTAATCGCAC-3') and a puc18 homology arm construct containing desired point mutations (R55A: ASP 3223, F43A: ASP 3222, or R55A/F43A: ASP 3543) with Fugene HD. After 4–5 days, GFP-positive cells were single-cell sorted into 96-well plates using a Sony SH800 cell sorter. Clonal populations were expanded and screened by western blotting for SUV39H1. Mutations in SUV39H1 were confirmed by purifying genomic DNA, PCR amplifying the targeted locus, and analyzing the amplicons by Sanger sequencing.

## Immunofluorescence

For mitotic spreads, cells were grown to about 70% confluency in T-25 or T-75 flasks, then mitotic cells were selected by shaking off less adherent cells. Released cells were pelleted, resuspended in 100 µL 75 mM KCl, then counted on a hemocytometer and diluted with more 75 mM KCl to $2 \times 10^5$ cells/mL. 100 µL cells were spun onto no. 1.5 glass coverslips using a Shandon Cytospin 4 at 2000 rpm for 5 min. Coverslips were hydrated with KCM buffer (10 mM Tris-HCl pH 8.0, 120 mM KCl, 20 mM NaCl, 0.5 mM EDTA, 0.1% Triton X-100), then cells were permeabilized for 10 min with KCM buffer with 0.5% Triton X-100. Coverslips were blocked with KCM + 2% BSA for 30 min, then stained with specific antibodies diluted in KCM + 2% BSA, 30 min each. After staining, cells were washed with KCM buffer and fixed with 3.7% formaldehyde in KCM for 10 min. After fixation, cells were washed with PBS, stained with 10 µg/mL Hoescht for 10 min, washed with PBS + 0.1% Triton X-100, washed with PBS, then mounted onto microscope slides.

For staining interphase DLD-1 cells, cells were plated on glass coverslips, fixed and permeabilized with 1% formaldehyde, 0.5% Triton X-100, 1X PBS for 10 min, then re-permeabilized with 0.5% Triton X-100, 1X PBS for 10 min. Cells were blocked in antibody dilution buffer (AbDil: 20 mM Tris-HCl pH 7.4, 150 mM NaCl, 0.1% Triton X-100, 2% BSA, 0.1% sodium azide) for 30 min, then stained with primary and secondary antibodies diluted in AbDil for 30 min each. Cells were then stained with 10 µg/mL Hoechst in Abdil for 10 min, washed with PBS + 0.2% Triton X-100, washed with PBS, then mounted onto slides and imaged.

We used the following antibodies for immunofluorescence: mouse anti-HEC1 (Abcam ab3613, 1:1000), rabbit anti-CENP-A-Alexa647 (Straight laboratory, 1:1000), rabbit anti-GFP-Alexa488 (Straight laboratory, 2 µg/mL), rabbit anti-HP1α (Straight laboratory, 2 µg/mL), rabbit anti-H3K9me3 (Abcam ab8898, 1:1000), human anti-centromere protein (CREST, Antibodies, inc., 1:100). All secondary antibodies were obtained from Invitrogen and diluted 1:1000. For experiments where

SUV39H1-GFP localization was assessed, expression of WT or mutant versions of SUV39H1-GFP was induced in HeLa Flp-In T-REx cells with 1 µg/mL doxycycline for 6 hr before mitotic shake off. To visualize RNA by EU labeling, cells were incubated with 0.5 mM EU (synthesized as previously described [*Jao and Salic, 2008*]) for 4–12 hr before mitotic shake off. In RNA polymerase inhibitor experiments, cells were incubated with 0.5 mM EU and 50 µg/mL α-amanitin (Santa Cruz Biotech.), 1 µM triptolide (Selleckchem), 50 ng/mL actinomycin D (Calbiochem), or 1 µM CX-5461 (Millipore) for 6 hr. Cells were processed and stained as above. After fixation, cells were washed with PBS, and EU-containing RNA was labeled using a Click-iT RNA Imaging kit (Invitrogen C10329), essentially according to the manufacturer's instructions. In RNase sensitivity experiments, 0.5 mg/mL RNase A (Sigma), III, or H was included in the 30 min block step before antibody staining (in KCM –EDTA + 5 mM MgCl$_2$ in experiments with RNase H and RNase III). Plus inhibitor (+inhibitor) controls also containing 4 U/µL each of RNaseOUT (Invitrogen) and RNase inhibitor (Ambion AM2684).

Stacks of fixed spread mitotic chromosomes and interphase nuclei were acquired at 0.2 µm axial steps using a motorized stage mounted on an Olympus IX70 microscope, using a 100X, 1.4 NA PlanApo objective and a CoolSnap-HQ CCD camera (Photometrics) on a DeltaVision Spectris system (Applied Precision). Displayed images are maximum intensity projections of raw or deconvolved z-stacks.

Fluorescent signal at pericentric regions of mitotic chromosomes or at centromeres in interphase nuclei was quantified using custom software. Pericentromere finder (*Fuller, 2014*): centromeres were localized using HEC1 staining, and pericentric regions were defined by segmenting on DNA on Hoescht-stained chromosomes within a 10-pixel radius around centromeres. Parameters used were min-size = 4, max-intercentromere-dist = 17, marker-channel = 2 (HEC1 staining), and pericentromere-channel = 0 (DNA staining). Centromere finder (*Fuller, 2016*): centromeres were localized using CREST staining, and fluorescence of each channel was measured. Parameters used were min_-size = 5, max_size = 60, marker_channel_index = 3 (CREST staining), decrease_speckle_background = false. Intensity measurements were performed on non-deconvolved maximum intensity projections of each z-series. GFP background signal was assessed by imaging spreads from cells that were not induced with doxycycline, and therefore were not expressing SUV39H1-GFP. Background in the HP1α channel was assessed by imaging spread cells not stained with HP1α primary antibody, and EU-RNA background was assessed with a no EU treatment control.

## RNA FISH

Mitotic DLD-1 cells were selected by mitotic shake off, washed with PBS, resuspended in 75 mM KCl to $5.8 \times 10^4$ cells/mL, then swelled at 37°C for 15 min. 500 µL cells were cytospun onto RNase AWAY treated slides (2000 rpm, 10 min). Cells were permeabilized in KCM buffer for 5 min, fixed in 4% PFA in PBS for 10 min, then blocked for 15 min (1X PBS, 0.5% Triton X-100, 1% BSA) and stained with specific antibodies (described above) in block. Cells were permeabilized again in CSK buffer/2 mM RVC(Ribonucleoside Vanadyl Complex)/0.5% Triton X-100, 10 min on ice, then fixed again in 4% PFA in PBS for 10 min. Slides were dehydrated with an ethanol series (70%, 80%, 95%, 100%, 2 min each on ice), then air-dried at 37°C. After precipitating and denaturing, probes were hybridized in hybridization mix (+RVC) at 37°C overnight in a humid chamber (pSD1-1 and ASP851 probes: 65% formamide; pβ4 and pTRS-47 probes: 50% formamide). Slides were then washed with 2X SSC/0.05% Tween/formamide (same formamide percentages used in hybridization mix), pH 7 at 37°C, 3 times 5 min each – then with 2X SSC/0.05% Tween, at 37°C, 3 times 5 min each – then 1X SSC/0.05% Tween, at room temp, 10 min – then 4X SSC/0.1% Tween, at room temp, 5 min. For biotin-16-dUTP labeled probes, after post-hybridization washes, slides were blocked in blocking buffer (PBS + 0.5% Triton X-100 + 1% BSA + 0.02% sodium azide) and biotin was detected with Alexa Fluor 488 streptavidin diluted in blocking buffer for 2 hr at room temperature. Slides were then washed with 4X SSC/0.1% Tween three times at room temp, 5 min each. All slides were stained with DAPI, coverslips were mounted, and images were acquired as described above. CSK buffer: 100 mM NaCl, 300 mM sucrose, 3 mM MgCl$_2$, 10 mM PIPES pH 6.8. Probe hybridization mix: 50% formamide (65–68% for repetitive), 10% dextran sulfate, 2X SSC, 1% Tween-20. RNase-treated controls slides were incubated with Ambion RNase cocktail (7.5 µL/mL KCM buffer) at 37°C for 2 hr up to overnight before hybridization.

DNA FISH was performed as above, with the following differences: slides were fixed in 10% formalin for 10 min at room temp, permeabilized in KCM for 10 min, denatured in 70% formamide, 2X SSC, pH 7 at 70°C for 3 min, then hybridized overnight.

All probes were labeled by nick translation of plasmids either indirectly with Roche biotin-16-dUTP and subsequently detected with AlexaFluor 488 streptavidin or directly with AlexaFluor 488-dUTP. D1Z5-specific RNA FISH probes were generated from the pSD1-1 plasmid, which contains a single copy of the 1.9 kb D1Z5 higher order repeat (HOR). Probes recognizing chromosome 13 and chromosome 21 α-satellite arrays were generated using a template PCR amplified from human genomic DNA with the following primers: α-satellite I (ASON 614): 5′-CTTGCTAGCAATCTGCAAG TGG-3′, and α-satellite II (ASON 615): 5′-CTTGTCGACTACAAAAAGAGTG-3′. β-satellite is detected with nick-translated plasmid pβ4, which recognizes β-satellite sequences on the acrocentric chromosomes (chromosomes 13, 14, 15, 21, and 22). Satellite III is detected with nick-translated plasmid pTRS-47, which recognizes Satellite III subfamily sequences on chromosomes 14 and 22. pSD1-1, pβ4, and pTRS-47 plasmids were a gift from the lab of Dr. Huntington Willard. 500 ng of 488 dUTP-labeled probes were precipitated with 0.5 uL salmon sperm DNA (10 mg/mL stock) and 2.5 volumes 100% EtOH for each $18 \times 18$ cm$^2$ area for at least 4 hr at $-80°$ C or overnight at $-20°$ C. Before use, probes were pelleted at maximum speed in an Eppendorf microcentrifuge, washed with 70% ethanol, resuspended in 65% formamide hybridization mix, and denatured at 74°C for 10 min.

The 'RGB profile plot' ImageJ plugin was used for line scan analysis of RNA FISH images.

## Protein expression

All purified SUV39H1 proteins used in this study were expressed as N-terminal Maltose Binding Protein (MBP) fusions. pMAL-c2X (NEB) expression vectors, carrying sequences encoding truncated versions of human SUV39H1, were transformed into BL21 *E. coli* and grown in 2–6 L of 2X YT at 37°C to an OD of 0.6–0.8. Cells were induced with 0.3 mM IPTG for 4–6 hr at room temperature. Cells were harvested, frozen in liquid nitrogen and stored at $-80°$C.

## Protein purification

Each liter of pelleted *E. coli* culture was resuspended in 20 mL of ice cold 4X lysis buffer (80 mM Tris-HCl pH 7.4, 0.8 M NaCl, 0.4% Triton X-100, 5 mM 2-mercaptoethanol, 1 mM PMSF, 1 mM benzamidine hydrochloride, 0.2 mg/ml lysozyme). Lysates were sonicated and centrifuged at 96,000Xg for 1 hr at 4°C. Supernatants were passed over 2–5 mL of either amylose resin (NEB) or glutathione agarose (Sigma Aldrich), followed by washing with approximately 10 column volumes of 4X lysis buffer, 30 column volumes of 1X lysis buffer (20 mM Tris-HCl pH 7.4, 0.2 M NaCl, 0.1% Triton X-100), and five column volumes of 1X lysis buffer without Triton X-100 at 4°C. Protein was eluted with approximately five col. volumes of 1X lysis buffer without Triton X-100 plus either 10 mM maltose or 5 mM glutathione. MBP fusions were dialyzed into 20 mM Tris-HCl pH 7.4, 50 mM NaCl, 10% glycerol, 5 mM 2-mercaptoethanol. For alanine scanning point mutants, frozen pellets from 100 mL cultures were thawed and resuspended in 5 mL of 4X lysis buffer. Lysates were sonicated, divided into $5 \times 1$ mL aliquots and spun at approximately 20,000Xg for 20 min at 4°C. Supernatants were combined and incubated with 500 μL of amylose resin (NEB) for 30 min at 4°C. The amylose resin was washed 3 times with 1 mL of 4X lysis buffer, 3 times with 1 mL of 1X lysis buffer without Triton X-100, and 3 times with 20 mM Tris-HCl pH 7.4, 50 mM NaCl, 10% glycerol, 5 mM 2-mercaptoethanol. Protein was eluted with 750 μL of 20 mM Tris-HCl pH 7.4, 50 mM NaCl, 10% glycerol, 5 mM 2-mercaptoethanol, 10 mM maltose for 30 min at 4°C. Amylose resin was pelleted by spinning at 20,000Xg for 2 min, and supernatants were collected. To remove contaminating RNases, proteins were loaded onto a 1 or 5 mL HiTrap SP sepharose FF column (GE) using an AKTA FPLC. The flow-through fraction was taken and concentrated using Amicon Ultra 30 kDa centrifugal filters (EMD Millipore). Protein concentrations were quantified by Bradford assay and $A_{280}$ measurement using a NanoDrop 1000 spectrophotometer. The absence of co-purifying nucleic acids was monitored by ensuring the absorbance ratio at 260 nm/280 nm was less than 0.7.

## Nucleic acid sequences used in EMSAs

RNA and DNA probes ≤50 bp were ordered from Integrated DNA Technologies.

Sense 19mer,

5'-AUAUGGGAACCACUGAUCC-3';
antisense 19mer,
5'-GGGAUCAGUGGUUCCCAUA-3';
sense MBP 1–50,
5'-ACCAAAAUCGAAGAAGGUAAACUGGUAAUCUGGAUUAACGGCGAUAAAGG-3';
antisense MBP 1–50,
5'-CCUUUAUCGCCGUUAAUCCAGAUUACCAGUUUACCUUCUUCGAUUUUGGU-3'.

To synthesize 180 bases of α-satellite or β-actin ssRNA, cDNAs were obtained from HeLa cell total RNA using the SuperScript III First-Strand Synthesis System (Life Technologies) according to the manufacturer's instructions, and the following primers:

α-satellite forward,
5'-CTTGCTAGCAATCTGCAAGTGG-3';
antisense β-actin,
5'-CGTAGATGGGCACAGTGTGG-3'.

cDNA was amplified into dsDNA using Phusion high-fidelity DNA polymerase (NEB) and the following primer pairs; α-satellite forward (sequence above) and

α-satellite reverse +T7,
5'- GTAATACGACTCACTATAGGGcttgtcgactacaaaaagagtg-3';
antisense β-actin (sequence above) and sense β-actin + T7,
5'- GTAATACGACTCACTATAGGGAGGCCCCCCTGAACCCCAAG-3'.

dsDNA was used for in vitro transcription as described below.

## Radiolabeling of nucleic acids

Nucleic acids were end-labeled using ATP $\gamma$-$^{32}$P (Perkin Elmer) and polynucleotide kinase (NEB). Approximately 15 pmol of ATP $\gamma$-$^{32}$P was added to 5–10 pmol of nucleic acid, polynucleotide kinase, and 10X reaction buffer, and incubated for 75 min at 37°C. The polynucleotide kinase was subsequently deactivated by incubating at 68°C for 20 min. Free ATP $\gamma$-$^{32}$P was separated from labeled nucleic acids using Micro Bio-spin Columns with Bio-Gel P-6 (Bio Rad) according to the manufacturers protocol, and exchanged into 20 mM Tris-HCl pH 7.4, 50 mM NaCl, 10% glycerol in DEPC $H_2O$. Labeled nucleic acids were diluted into approximately 1.5 mLs of Tris-HCl pH 7.4, 50 mM NaCl, 10% glycerol in DEPC $H_2O$ to a final concentration of approximately 1–5 nM. MBP 1–50 nucleic acids were further purified by native gel electrophoresis (13.33% acrylamide 29:1, 0.5X TBE). Bands were identified using HyBlot CL Autoradiography Film (Denville Scientific), excised and eluted for 3 hr at 25°C into 1 mL of 20 mM Tris-HCl, pH 7.4, 50 mM NaCl, 10% glycerol in DEPC $H_2O$ at a final concentration of ≤1 nM. α-satellite and β-actin ssRNAs were further purified by denaturing gel electrophoresis (10% urea, 10% acrylamide 29:1, 1X TBE). Bands were identified using HyBlot CL Autoradiography Film (Denville Scientific), excised and eluted for 3 hr at 25°C into 1 ml DEPC $H_2O$ with the addition of 160 units of RNase Inhibitor (Ambion, AM2684). RNA was concentrated using Amicon Ultra 30 kDa centrifugal filters and washed 3 times with 500 µL DEPC $H_2O$. RNA was resuspended in 1 mL of 20 mM Tris-HCl pH 7.4, 50 mM NaCl, 10% glycerol, made with DEPC-treated water at a final concentration approximately ≤1 nM.

## In vitro transcription

RNA was transcribed from α-satellite and β-actin dsDNA (from cDNA described in previous section, nucleic acid sequences used in EMSAs) using the MEGAscript T7 Transcription Kit (Life Technologies) according to the manufacturers protocol, with the addition of 40 units of RNase Inhibitor (Ambion, AM2684). Transcribed RNA was loaded onto a 1% agarose gel to verify lengths, and RNA concentrations were measured using a NanoDrop 1000 spectrophotometer. Approximately 2 µg of RNA was desphosphorylated for 60 min at 37°C using Antarctic Phosphatase (NEB) with the addition of 40 units of RNase Inhibitor (Ambion, AM2684) in preparation for end-labeled using ATP $\gamma$-$^{32}$P.

## Electrophoretic mobility shift assays (EMSAs)

Purified proteins and nucleic acids were thawed at room temperature and kept on ice. Proteins were serially diluted into binding buffer (20 mM Tris-HCl pH 7.4, 50 mM NaCl, 10% glycerol in DEPC $H_2O$) and kept on ice until mixed with RNA. Proteins were mixed with $^{32}$P-labeled nucleic acids and

approximately 4 units RNase Inhibitor (Ambion, AM2684), and allowed to equilibrate to at room temperature for 10 min before loading onto a native gel (7.5–13.33% acrylamide 29:1, 0.5X TBE) buffered with 0.5X TBE. Gel electrophoresis was carried out for 1 hr at 25°C at 10 mAmps. Gels were exposed to phosphorimaging screens for approximately 1 hr before signal acquisition was performed using a Typhoon 9400 Variable Mode Imager (GE). Gels were quantified using FIJI software, and binding curves, dissociation constants, and standard error were plotted and calculated using Graphpad Prism software.

## Alanine scanning mutagenesis

Individual alanine point mutants were synthesized using PCR assembly of six overlapping DNA oligonucleotides (Integrated DNA Technologies) of approximately 60 bp, spanning a 5′ AscI site, the SUV39H1 chromodomain (residues 42–106), and a 3′ PacI site. Following PCR assembly, DNA was purified using Agencourt AMPure XP beads. 40 µL of each PCR product was mixed with 72 µL Ampure beads and incubated for 5 min at 25°C. Beads were collected using a 24-well magnetic stand for 5 min, and washed twice with 200 µL 70% ethanol. Beads were air-dried for 20 min at 25°C, and DNA was eluted with 40 µL of $H_2O$ for 5 min at 25°C. Recovered DNA was quantified using a NanoDrop 1000 spectrophotometer. Each point mutant was cloned into the AscI/PacI sites of the pMAL-c2X vector and verified by sequencing.

## Filter binding assays

Purified proteins and nucleic acids were thawed at room temperature and kept on ice. Proteins were serially diluted into binding buffer (20 mM Tris-HCl pH 7.4, 50 mM NaCl, 10% glycerol in DEPC $H_2O$) and kept on ice until mixed with RNA. Proteins were mixed with $^{32}$P-labeled sense 19mer RNA and approximately 4 units RNase Inhibitor (Ambion, AM2684), and allowed to equilibrate to at room temperature for 10 min. 4 µL of each binding reaction was dotted onto a nitrocellulose membrane. The membrane was washed several times with binding buffer without glycerol, and membranes were exposed to phosphorimaging screens and imaged using a Typhoon 9400 Variable Mode Imager (GE). Gels were quantified using FIJI software, and binding curves, dissociation constants, and standard error were plotted and calculated using Graphpad Prism.

## In vitro peptide binding assays

SUV39H1 42–106 proteins were diluted to a final concentration of 12.5 µM in 20 mM Tris-HCl pH 7.4, 50 mM NaCl, 10% glycerol. Histone H3 1–20 peptides (EpiCypher) were diluted to 40 µM in peptide binding buffer (50 mM Tris-HCl pH 7.4, 150 mM NaCl, 0.05% NP-40, 10 mg/mL BSA). 5 µL of SUV39H1 42–106 (1.25 µM final concentration) was mixed with 1 µL peptide (800 nM final concentration) and 44 µL peptide binding buffer and incubated for 45 min at 4°C. 20 µL of Dynabeads M-280 Strepavidin (Life Technologies) were washed 3 times with 1 mL of peptide binding buffer, added to the binding reaction and incubated for 45 min at 4°C. Beads were washed 4 times with 1 mL of peptide binding buffer and eluted with 100 µL of protein SDS sample buffer (10% SDS, 50% glycerol, 0.05% bromophenol blue, 0.2 M Tris-HCl pH 6.8, 40 mM EDTA pH 8, 20% 2-mercaptoethanol). 20 µL was loaded onto a 12.5% polyacrylamide SDS denaturing gel and transferred onto a PVDF membrane for western blotting with anti-MBP antibody. Bands were quantified using FIJI software and p-values for an unpaired t-test were calculated using Graphpad Prism software.

## Structure comparison

The structures of the chromodomains of human HP1α bound to H3K9me3 (*Kaustov et al., 2011*) (PDB ID:3FDT) and human SUV39H1 (*Wang et al., 2012*) (PDB ID:3MTS) were aligned using the Matchmaker Tool in the Chimera Software Package (*Pettersen et al., 2004*). Residue F43 of SUV39H1 is not shown in the comparison because it is not present in the crystal structure.

## Histone methyltransferase activity assay

150 µg/mL full-length SUV39H1 proteins were incubated with 70 µg/mL Xenopus H3/H4 tetramer (*Guse et al., 2012*), 0.8 µCi/mL $^{14}$C-SAM (Perkin Elmer), ±124 µg/mL β-actin RNA (360 bp, cDNA amplified with ASON 2584/2582 and reverse transcribed), in activity buffer (10 mM HEPES pH 7.7, 100 mM KCl, 1 mM $MgCl_2$, 1 mM $CaCl_2$, 0.5 mM DTT) in a final volume of 20 µL. Reactions

proceeded at 30°C for the indicated time, then samples were boiled in SDS-PAGE loading buffer for 5 min and saved at −20°C. Samples were run on a 20% SDS-PAGE gel; the gel was fixed, Coomassie stained, and dried; and methylation was detected with a phosphorimaging screen scanned on a Typhoon 9400 Variable Mode Imager (GE).

## Western blotting

Rabbit anti-MBP, anti-GFP, and anti-HP1α polyclonal antibodies were generated by Cocalico Biologicals and purified from rabbit serum. All primary antibodies were incubated for 1 hr (except for anti-SUV39H1 which was incubated overnight) at the following dilutions: rabbit anti-MBP (Straight lab, 0.2 µg/mL), rabbit anti-GFP (Straight lab, 1 µg/mL), rabbit anti-HP1α (Straight lab, 1 µg/mL), mouse anti-SUV39H1 (Millipore clone MG44, cat. #05–615, 1:500), rabbit anti-H3K9me3 (AbCam ab8898, 1:1000), rabbit anti-histone H4 (AbCam ab7311, 1:2000), and mouse monoclonal anti-tubulin (clone DM1α, Sigma T6199, 0.5 µg/mL).

To western blot for endogenous SUV39H1 in DLD-1 cells, nuclear lysates were prepared to enrich for the SUV39H1 containing fraction. Cells were incubated in hypotonic buffer (10 mM Tris-HCl pH 8, 1.5 mM MgCl2, 10 mM KCl) for 10 min on ice, dounced 10 times with a Kontes glass dounce (B pestle), nuclei were pelleted at 5000xg for 10 min, then lysed in DLB (*Singh et al., 2014*). If necessary, lysates were sonicated with a Branson Sonifier S-250A using a microtip until no longer viscous. To test the effect of RNase A on SUV39H1 protein levels, pelleted nuclei were incubated with 0.5 mg/mL RNase A, ±4 U/µL RNaseOUT, for 30 min before lysis.

## RNA immunoprecipitation (RIP) assay

All buffers listed here are described in *Singh et al. (2014)*. HeLa Flp-In T-REx cell lines expressing GFP, WT SUV39H1-GFP, or R55A SUV39H1-GFP were expanded to six 15 cm dishes each, and protein expression was induced with 1 µg/mL doxycycline for 6 hr. Cells were trypsinized, pelleted, and then resuspended in 30 mL PBS. To crosslink cells, formaldehyde was added to a final concentration of 0.1%, and cells were rotated at room temperature for 10 min. Crosslinking was stopped by adding 3 mL quenching buffer, then rotating cells for 5 min. Cells were pelleted and lysed in 3 mL denaturing lysis buffer (DLB) on ice for 10 min. Lysates were sonicated with a Branson Sonifier S-250A using a microtip (setting 4.5, 2 rounds of 10 pulses), diluted up to 8 mL with DLB, then cleared by spinning at 12,800 rpm then 33,000 rpm in a Type 70.1 Ti rotor for 10 min each at 4°C. Lysates were all diluted to 0.3 mg/mL, 20 mL each, and samples were saved for western blotting and analysis of total input RNA. Remaining lysate was added to 40 µL NHS-activated sepharose coupled to a llama anti-GFP nanobody (*Rothbauer et al., 2008*) and rotated at 4°C for 2 hr. Beads were collected by brief centrifugation, and supernatant sample was saved for western analysis. Beads were washed 2 × 10 mL with denaturing wash buffer (DWB), 2 × 10 mL with isotonic wash buffer (IsoWB), then resuspended in 100 µL of clear sample buffer (CSB). Crosslinks were reversed in both input lysate and IP samples by heating at 75°C for 40 min. 5 µL of each IP supernatant was saved for western blotting and silver staining, and the remaining supernatant and input lysates were mixed with 1 mL TriPure isolation reagent (Roche). RNA was purified using TriPure, then treated with 10 units of TURBO DNase (Life Technologies) at 37°C overnight according to the manufacturer's instructions. RNA was re-purified with TriPure, resuspended in 20 µL DEPC-treated water, then analyzed by RT-qPCR. Silver staining of immunoprecipitated material was performed essentially as described (*Chevallet et al., 2006*).

## RT-qPCR

RNA was reversed transcribed into cDNA using the SuperScript III First-Strand Synthesis System (Life Technologies 18080–051) per manufacturer's instructions, using the following RT primers: α-satellite I (ASON 614): 5'-CTTGCTAGCAATCTGCAAGTGG-3', β-actin rev (ASON 2430): 5'-ATGTCCACG TCACACTTCAT-3', and GAPDH rev: 5'-CCTGCTTCACCACCTTCTT-3'. Quantitative PCR (qPCR) was then performed with the same reverse primers plus the following forward primers: α-satellite II (ASON 615): 5'-CTTGTCGACTACAAAAAGAGTG-3', β-actin fwd (ASON 2392): 5'-AGAGCTAC-GAGCTGCCTGAC-3', and GAPDH fwd (ASON): 5'-AGATCATCAGCAATGCCTCC-3'. qPCR mix recipe: 1X Phusion HF buffer (NEB), 3% DMSO, 200 µM dNTPs, 1X ROX reference dye (Life Technologies), 1X SYBR green I (Invitrogen S-7563), Phusion polymerase, and 100 nM each primer. The following

programs were used: α-satellite (98℃ 30 s, then 40X cycles of 98℃ 15 s, 60℃ 15 s, 72℃ 15 s), *β*-actin (98℃ 30 s, then 40X cycles of 98℃ 10 s, 72℃ 30 s). qPCR was performed on a 7900HT Fast Real-Time PCR System (Applied Biosystems), and the number of cycles to reach the automatically set threshold ($C_t$) were determined using Sequence Detection Systems software. For each primer set, standard curves of diluted cDNA were used to assess the exponential amplification (expAmp) of the amplicon. The signal in each experimental sample was determined by signal = expAmp$^{-Ct}$. For all α-satellite measurements, '-RT' signal was subtracted from '+RT' signal, as the repetitive nature of α-satellite DNA leads to high background. α-satellite signal was then divided by the *β*-actin or GAPDH signal to get final α-satellite normalized signal. For IP enrichment values, the α-satellite normalized signal for the IP was then divided by the α-satellite normalized signal of the input lysate.

To test inhibition of transcription after adding RNA polymerase inhibitors, RT-qPCR was performed essentially as described above with the following primers:

pre-tRNA(tyr) fwd (ASON 1991): CCTTCGATAGCTCAGCTGGTAGAG
pre-tRNA(tyr) rev (ASON 1990): AAAAAACCGCACTTGTCTCCTTCG
GAPDH pre-mRNA fwd (ASON 4619): CATGCCTTCTTGCCTCTTGT
GAPDH pre-mRNA rev (ASON 4620): TGAGGTCAATGAAGGGGTCA
45S pre-rRNA fwd (ASON 1891): CCTGCTGTTCTCTCGCGCGTCCGAG
45S pre-rRNA rev (ASON 1892): AACGCCTGACACGCACGGCACGGAG

## RNA-seq and analysis of repeat families

For total RNA-seq analysis of control and SUV39 DKO HeLa and DLD-1 cells, total RNA was purified using TriPure, then treated with 10 units of TURBO DNase (Life Technologies) at 37℃ for 30 min per the manufacturer's instructions. RNA was re-purified with TriPure, resuspended in DEPC-treated water, and quality checked by Bioanalyzer (Agilent). RNA was then analyzed either by RT-qPCR to measure α-satellite RNA levels as described above, or a cDNA library was generated for RNA-Seq analysis. For RNA-Seq, ribosomal RNAs were first depleted using a NEBnext rRNA depletion kit (New England Biolabs), RNA was purified using Agencourt Ampure XP beads (Beckman Coulter), then cDNAs were generated, amplified, and indexed with the ScriptSeq v2 RNA-Seq Library Preparation Kit (Epicentre) per the manufacturer's instructions. Indexed libraries were quantified by Bioanalyzer and qPCR, pooled, and sequenced on a NextSeq 500 (Illumina).

For analysis of repetitive RNAs, raw fastq files were adapter trimmed and converted to fasta format. The fasta files were then split into smaller files of 1,000,000 sequences each. All smaller fasta files of each original fasta file were analyzed by RepeatMasker 4.0.3 using the human species database (*Bao et al., 2015*). The RepeatMasker output file for each smaller fasta file was merged together to give the repeat content of the original fasta file. Repeat class and type was summarized using the buildSummary.pl script in the RepeatMasker utility script folder. The number of reads for each repeat in the RepeatMasker output file was normalized to the total number of repeats detected by RepeatMasker to obtain a frequency of detection for each repeat type. To calculate the fold enrichment for each repeat type, the frequency of each read in the SUV39 DKO cell line was divided by the frequency of the same read in the control cell line. A cutoff threshold for inclusion in the analysis was set at 300 reads per repeat type.

Analysis of non-repetitive RNA was performed by aligning reads (after PCR duplicate removal and adaptor trimming) to the UCSC hg38 build (downloaded Aug 14, 2015 and archived by Illumina iGenomes) using tophat2 (v 2.2.9). The aligned reads were annotated and quantified using Cufflinks (v 2.2.1).

## Fluorescence recovery after photobleaching (FRAP)

Flp-In HeLa cells containing WT or mutant SUV39H1-GFP were grown on glass-bottom chambered coverglass (LabTek) under standard conditions. SUV39H1-GFP expression was induced by adding 1 µg/mL doxycycline six hours before data collection. For drug treatment experiments, cells were treated with 1 µM CX-5461 (Millipore), 50 µg/mL α-amanitin (Santa Cruz Biotech), or 1 µM triptolide (Selleckchem) for 2 hr before image acquisition. Triptolide and α-amanitin experiments were done in cells induced with 50 ng/mL doxycycline for 24 hr. Immediately before the experiment, the media was exchanged to pH-indicator free media (DMEM, 10% FBS, constant doxycycline and drug concentration). Microscopy was performed with a Nikon Eclipse Ti at 100x magnification.

Photobleaching was performed using an Andor Mosaic II patterned illumination system coupled to a 450 mW 405 nm laser. Photobleaching was performed using 100% power for one second using a circular spot with a diameter of ~3 μm. Images of the recovery were collected, and intensity profiles of the bleaching area and of the whole nucleus were calculated using ImageJ 2.0. Intensity profiles were normalized using the following formula:

$$I = \frac{ROI_{(t)}}{ROI_{(0)}} \cdot \frac{total_{(t)}}{total_{(0)}} \tag{1}$$

$$I_{norm} = \frac{I_{(t)} - I_{(bleach)}}{I_{(0)} - I_{(bleach)}} \tag{2}$$

Where ROI and total are the intensities of the photobleaching area and whole nucleus, respectively, at either time zero or at time $t$. The intensity at time zero is $I_{(0)}$ (i.e., before bleaching), $I_{(bleach)}$ is the intensity immediately after bleaching, and $I_{(t)}$ is the intensity at subsequent time intervals. The normalized intensity traces were then analyzed for each cell using the following single-exponential function:

$$Y = Y_{max} \cdot \left( 1 - e^{\left( \frac{-ln2}{t_{1/2}} \right)(x - x_0)} \right) \tag{3}$$

where $Y_{max}$ is the plateau of the recovery curve and $t_{1/2}$ is the halftime for recovery, in seconds. The amplitude of the plateau is equivalent to the fraction of mobile protein.

## Acknowledgements

This work was funded by the American Cancer Society Research Scholar Grant (12061-RSG-11-025-01-CCG) to AFS. WLJ was supported by a NIH T32 Training Fellowship (GM007276), the National Science Foundation Graduate Research Fellowship (DGE-114747), and the Stanford Graduate Fellowship Program. WTY was supported by a NIH T32 Training Fellowship (GM007276) and the National Science Foundation Graduate Research Fellowship (DGE-114747), JCB was supported by the Stanford School of Medicine Dean's Postdoctoral Fellowship and the NIH F32 Ruth L. Kirschstein National Research Service Award (GM116378). SMM was supported by the National Science Foundation Graduate Research Fellowship (DGE-1644868), BAS was supported by NIH R01 GM098500 and March of Dimes grant 1-FY13-517. RJO and ZD were supported by NSF grant number 121132. Any opinion, findings, and conclusions or recommendations expressed in this material are those of the authors and do not necessarily reflect the views of the National Science Foundation. We thank members of the Straight lab for support, helpful feedback, and critical reading of the manuscript. We thank Dr. Colin J. Fuller for writing custom software. We thank Siqi Tian in Dr. Rhiju Das' lab for his assistance in the PCR assembly of SUV39H1 alanine scanning mutants. We thank Dr. Daniel Herschlag for helpful input on binding experiments. We thank Dr. Mark Smith of the Stanford ChEM-H Medicinal Chemistry Knowledge Center for his assistance in synthesizing and purifying EU.

## Additional information

### Funding

| Funder | Grant reference number | Author |
|---|---|---|
| National Institutes of Health | GM007276 | Whitney L Johnson<br>William T Yewdell |
| National Science Foundation | DGE-114747 | Whitney L Johnson |
| Stanford University School of Medicine | Stanford Graduate Fellowship Program | Whitney L Johnson |
| National Science Foundation | DGE-114747 | William T Yewdell |
| National Institutes of Health | GM116378 | Jason C Bell |

| | | |
|---|---|---|
| Stanford University School of Medicine | Dean's Postdoctoral Fellowship | Jason C Bell |
| National Science Foundation | DGE-1644868 | Shannon M McNulty |
| National Science Foundation | 121132 | Zachary Duda<br>Rachel J O'Neill |
| National Institutes of Health | GM098500 | Beth A Sullivan |
| March of Dimes Foundation | 1-FY13-517 | Beth A Sullivan |
| American Cancer Society | 12061-RSG-11-025-01-CCG | Aaron F Straight |

The funders had no role in study design, data collection and interpretation, or the decision to submit the work for publication.

### Author contributions

WLJ, Conceptualization, Formal analysis, Validation, Investigation, Methodology, Writing—original draft, Writing—review and editing; WTY, Conceptualization, Formal analysis, Validation, Investigation, Visualization, Methodology, Writing—original draft, Writing—review and editing; JCB, Formal analysis, Validation, Investigation, Visualization, Methodology, Writing—original draft, Writing—review and editing; SMM, Validation, Investigation, Visualization, Methodology, Writing—review and editing; ZD, Software, Formal analysis, Methodology; RJO'N, Software, Formal analysis, Supervision, Funding acquisition, Validation, Investigation, Visualization, Writing—review and editing; BAS, Formal analysis, Funding acquisition, Validation, Investigation, Visualization, Methodology, Writing—review and editing; AFS, Conceptualization, Formal analysis, Supervision, Funding acquisition, Validation, Investigation, Visualization, Methodology, Writing—original draft, Project administration, Writing—review and editing

### Author ORCIDs

Whitney L Johnson, http://orcid.org/0000-0002-1442-3309
William T Yewdell, http://orcid.org/0000-0002-5712-1896
Jason C Bell, http://orcid.org/0000-0001-5480-7975
Beth A Sullivan, http://orcid.org/0000-0001-5216-4603
Aaron F Straight, http://orcid.org/0000-0001-5885-7881

# Additional files

### Major datasets

The following datasets were generated:

| Author(s) | Year | Dataset title | Dataset URL | Database, license, and accessibility information |
|---|---|---|---|---|
| Johnson W, Duda Z, O'Neill R, Straight A | 2017 | RNA Seq in control HeLa cells | http://www.ncbi.nlm.nih.gov/biosample/6271914 | Publicly available at the NCBI BioSample (accession no: SAMN06271914) |
| Johnson W, Duda Z, O'Neill R, Straight A | 2017 | RNA Seq in SUV39H1/H2 knockout HeLa cells | http://www.ncbi.nlm.nih.gov/biosample/6271915 | Publicly available at the NCBI BioSample (accession no: SAMN06271915) |

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
