## [Decision Letter]

Thank you for submitting your article "RNA-dependent stabilization of SUV39H1 at constitutive heterochromatin" for consideration by *eLife*. Your article has been reviewed by 3 peer reviewers, and the evaluation has been overseen by a Reviewing Editor and Jessica Tyler as the Senior Editor. The reviewers have opted to remain anonymous.

The reviewers have discussed the reviews with one another and the Reviewing Editor has drafted this decision to help you prepare a revised submission.

Summary:

This study explores the role of RNA in localization of the human SUV39H1 to chromatin. The authors perform a careful series of experiments showing that the stable association of a moderately overexpressed SUV39H1-EGFP protein with heterochromatic foci is RNase-sensitive. The authors further show that major satellite repeats are transcribed and give rise to RNAs that remain associated with mitotic chromosomes and are present in SUV39H1 immunoprecipitates. Using purified SUV39H1, the authors demonstrate that the enzyme binds to single stranded RNA of any sequence and identify specific mutations in the chromo domain of SUV39H1 that disrupt RNA binding without affecting its interaction with H3K9me3 peptides.

Finally, the authors demonstrate that the RNA binding mutations affect the stable binding of SUV39H1 to chromatin in vivo. Importantly, when the chromo domain RNA binding defective mutants are introduced into knockout cells, their weak association with centromeric heterochromatin is not further reduced by RNase treatment, providing strong support for the idea that specific interactions between the SUV39H1 chromo domain and RNA stabilize its association with heterochromatin. The experiments in the paper are straightforward and support the main conclusion of the paper. The model that the authors present, suggesting that non-specific RNA binding works alongside H3K9me3 to stabilize the association of SUV39h1 with chromatin, is reasonable and potentially important.

However, what is lacking in the study is any demonstration that the RNA binding activity described by authors contributes to the silencing function of SUV39H1 in vivo. This seems like a major shortcoming that needs to be addressed prior to publication.

Essential revisions:

1) A demonstration of functional importance of the RNA binding needs to be performed. This should be tested by introducing wildtype versus RNA-binding mutant SUV39H1 into their SUV39H1/H2 DKO cells and seeing what happens to H3K9me levels and what happens to Silencing of major satellite repeats and other transposons?

2) The authors make the intriguing observation that transcription of α satellite RNA is not eliminated by inhibition of RNA pol II, but is eliminated by the inhibition of both RNA pol I and pol II (actinomycine D treatment). It would be very surprising if this were in fact the case. Does pol I localize to α repeats (by ChIP)? I think that the more likely explanation might be that their α-amanitin treatment is not inhibiting pol II completely. You need to show data that more convincingly clarify whether only PolII transcription is occurring, or better evidence for Pol I transcription at the α satellites.

3) Figure 4 – you need to state the binding affinity of R55A & R84A mutants for nucleic acid substrates used in Figure 3 to determine whether the decrease in binding is specific to RNA substrates. Without these data the assertion that R55A and R84A mutants are defective for binding to RNA only is untested in vitro, and would need to be removed from the text.

4) Comparing Figure 4 and Figure 5: Is the decrease in R55A's ability to bind to α-satellite RNA due to a decrease in H3K9me binding or interaction with RNA? The in vitro binding experiments (4E, F) reveal a decrease in H3K9me binding for R55A compared to the wild-type protein. This can contribute to the in vivo decrease in RIP signal in Figure 5. This should be ruled out experimentally (see below).

5) A prediction of the proposed model is that loss of CD-mediated nucleic acid and H3K9me binding together should reduce SUV39H1 interaction with α-satellite RNAs more than either mutation alone. This can be done by creating a protein carrying a nucleic acid-binding mutation (R55A/R84A) with an H3K9me-binding mutation (W64/F43) (or a SET domain mutation in a Suv39H1/H2 mutant cell line). RIP experiments with these mutants will permit the authors to test this prediction directly. Either these experiments should be done or the model toned down.

6) Figure 2 and 6. Please show data to demonstrate that RNase A treatment of cells does not cause a decrease in SUV39H1 protein levels.

Other revisions:

1) Figure 1 and Figure 1—figure supplement 1 – why only 4 out of the 8 cell lines tested show α-satellite pericentric RNA localization in mitosis? It will be good to see RNA-FISH data for HeLa cells as well.

2) For Figure 3 – what is the affinity of NTE (1-42 aa's), for different nucleotide substrates? It will be good control to understand if NTE, by itself, has any nucleotide binding capacity.

3)Subsection “RNA associates with the pericentric regions of human mitotic chromosomes”, last paragraph: Are there other genomic regions where the RNA is associated with mitotic chromosomes in cis? If yes, then what are those euchromatic/heterochromatic regions?

4) Figure 6 – is the methylation efficiency of R55A and R84A mutants comparable to that of wild type protein? This will also impact results presented in Figure 6—figure supplement 1.

5) Subsection “A model for the RNA-dependent stabilization of SUV39H1 on chromatin”, first paragraph. The discussion of the pombe HP1/Swi6 and its requirement for Clr4-dependent H3K9me is incorrect. Even though this was shown in the referenced paper, subsequent work using the complete deletion of swi6 (instead of a this special swi6 mutant) have shown that loss of Swi6 has little impact on Clr4-dependent H3K9me (Sadaie et al. 2004 EMBO; Motamedi et al. 2008 Mol Cell).

6) Figure 6—figure supplement 1. Please show error bars for wt and calculate p values for these data.

7) Subsection “SUV39H1 depends on direct RNA binding for its stable association with heterochromatin”, first paragraph. Please keep mobile/immobile percentages consistent between wt and mutant proteins. For example, convert 42% immobile fraction to 58% mobile considering the mobile fraction of mutants is indicated.

---

## [Author Response]

*Essential revisions:*

*1) A demonstration of functional importance of the RNA binding needs to be performed. This should be tested by introducing wildtype versus RNA-binding mutant SUV39H1 into their SUV39H1/H2 DKO cells and seeing what happens to H3K9me levels and what happens to Silencing of major satellite repeats and other transposons?*

We agree with this suggestion, and have now assessed the functions of the SUV39H1 nucliec acid binding mutants on the levels of H3K9me3 and on transcription of α-satellite sequences. To do this, we used CRISPR/Cas9 to make individual nucleic acid binding and H3K9me3 binding mutations in the endogenous SUV39H1 gene. We performed this gene editing in SUV39H2 knockout cells so that the activities of SUV39H2 did not obscure the analysis of mutant phenotypes. The results, shown in the new Figure 8, demonstrate that both nucleic acid binding and H3K9me3 binding by SUV39H1 contribute to α-satellite repression, and that ablating both interactions is largely equivalent to removing SUV39H1 altogether (Figure 2—figure supplement 1). The loss of repression of α-satellite sequences is accompanied by measurable decreases in H3K9me3 methylation. Interestingly, the R55A/F43A double mutant displays no additional defect in H3K9me3 compared to F43A alone, even though it shows a significant increase in α-satellite RNA levels – suggesting RNA binding of SUV39H1 may contribute to H3K9 methylation-independent roles for SUV39H1 in transcriptional repression. We believe this new data convincingly shows the functional importance of RNA binding by SUV39H1 and its importance in maintaining silencing of constitutive heterochromatin, and significantly strengthens the manuscript.

*2) The authors make the intriguing observation that transcription of α satellite RNA is not eliminated by inhibition of RNA pol II, but is eliminated by the inhibition of both RNA pol I and pol II (actinomycine D treatment). It would be very surprising if this were in fact the case. Does pol I localize to α repeats (by ChIP)? I think that the more likely explanation might be that their α-amanitin treatment is not inhibiting pol II completely. You need to show data that more convincingly clarify whether only PolII transcription is occurring, or better evidence for Pol I transcription at the α satellites.*

We agree that incomplete Pol II inhibition could explain the EU-RNA signal on chromosomes after α-amanitin treatment. Because we were unable to find a Pol I antibody suitable for ChIP, we addressed this issue by performing a new set of experiments using a more diverse panel of RNA polymerase inhibitors – including triptolide, that exhibits much faster Pol II inhibition kinetics than α-amanitin (Bensaude, 2011) – and by providing a more quantitative assessment of EU-RNA levels and RNA polymerase inhibition specificity (new Figure 1—figure supplement 2). We find that both Pol II inhibitors (α-amanitin and triptolide) decrease EU-RNA levels by about 50%; however, our RT-qPCR analysis shows that EU-RNA levels largely correlate with Pol I inhibition, and although actinomycin D and CX-5461 show no inhibition of Pol II, they reduce EU-RNA down to background levels. This strongly indicates that the EU-RNA signal on mitotic chromosomes is predominantly transcribed by Pol I. The EU-RNA signal we observe likely includes other RNAs in addition to α- satellite. To examine α-satellite transcripts directly, we measured their levels by RT-qPCR and observed that none of our polymerase inhibition conditions led to a decrease in total α-satellite (although triptolide treatment resulted in a substantial α-satellite RNA increase) (Figure 1—figure supplement 2). This suggests that the α-satellite transcribed during our 6 hr treatment is likely a small subset of total α-satellite.

*3) Figure 4 – you need to state the binding affinity of R55A & R84A mutants for nucleic acid substrates used in Figure 3 to determine whether the decrease in binding is specific to RNA substrates. Without these data the assertion that R55A and R84A mutants are defective for binding to RNA only is untested in vitro, and would need to be removed from the text.*

We agree that the R55A and R84A mutations likely disrupt binding to all nucleic acid substrates, including RNA and DNA. Therefore, we have changed our language in the manuscript to refer to these as nucleic acid binding mutants. One important note is that the treatment of mitotic chromosomes with RNase A, or treatment of interphase cells with RNA polymerase inhibitors, recapitulates defects of the nucleic acid binding mutants in these assays, indicating that the relevant binding substrate for SUV39H1 localization on chromosomes is likely RNA. Thus, in cases where we can ascribe the primary defect we observe in the mutants to RNA binding, we have maintained this phrase.

*4) Comparing Figure 4 and Figure 5: Is the decrease in R55A's ability to bind to α-satellite RNA due to a decrease in H3K9me binding or interaction with RNA? The in vitro binding experiments (4E, F) reveal a decrease in H3K9me binding for R55A compared to the wild-type protein. This can contribute to the in vivo decrease in RIP signal in Figure 5. This should be ruled out experimentally (see below).*

We acknowledge that the R55A mutation appears to cause a slight but insignificant decrease in SUV39H1 binding to H3K9me3. However, if the R55A mutation were really disrupting H3K9me3 binding, we would expect this to have a measurable effect on SUV39H1 localization. This is clearly not the case, as the localization defects of the R55A mutant – observed by IF on mitotic chromosomes and by FRAP in interphase cells – quantitatively matches the localization defects of R84A (Figure 6, Figure 7), which exhibits no measurable defect in H3K9me3 binding (Figure 4). Therefore, we can be confident that the RIP difference we observe is due to R55A’s effects on nucleic acid binding, not on H3K9me3 binding (see also the response to point 5).

*5) A prediction of the proposed model is that loss of CD-mediated nucleic acid and H3K9me binding together should reduce SUV39H1 interaction with α-satellite RNAs more than either mutation alone. This can be done by creating a protein carrying a nucleic acid-binding mutation (R55A/R84A) with an H3K9me-binding mutation (W64/F43) (or a SET domain mutation in a Suv39H1/H2 mutant cell line). RIP experiments with these mutants will permit the authors to test this prediction directly. Either these experiments should be done or the model toned down.*

We agree that a prediction of our model is that breaking nucleic acid binding and H3K9me3 binding together should have a larger effect on SUV39H1 function than breaking either of these interactions in isolation. To test this, we have performed new experiments (new Figure 8) that demonstrate that a double mutant of SUV39H1 that disrupts both nucleic acid binding and H3K9me binding (R55A/F43A) causes more substantial derepression of α-satellite transcription than either individual mutant alone (Figure 8). This experiment also addresses the above concern (point 4) – that effects of the R55A mutant may be due to a slight decrease in H3K9me3 recognition (Figure 4) – in that an R55A mutation would not add an additional defect on top of that contributed by F43A if R55A were solely acting through disrupting H3K9me3 binding.

*6) Figure 2 and 6. Please show data to demonstrate that RNase A treatment of cells does not cause a decrease in SUV39H1 protein levels.*

We have included a new supplemental figure (Figure 2—figure supplement 1) showing that RNase treatment does not decrease SUV39H1 protein levels.

Other revisions:

*1) Figure 1 and Figure 1—figure supplement 1 – why only 4 out of the 8 cell lines tested show α-satellite pericentric RNA localization in mitosis? It will be good to see RNA-FISH data for HeLa cells as well.*

We have now included data showing α-satellite RNA FISH signal on HeLa mitotic chromosomes (new Figure 1—figure supplement 1). Potential explanations for why we do not detect pericentric EU-RNA in all cells are: 1) RNA FISH is a more sensitive assay than EU-RNA labeling, EU is incorporated in only a subset of uridine positions and it only labels RNA transcribed during our incubation period, 2) there may be variation from cell line to cell line how much RNA is bound to chromosomes during mitosis. Because we can detect pericentric α- satellite RNA by RNA-FISH in both HeLas and DLD-1s, but only detect pericentric EU-RNA in HeLas, we conclude that the RNA FISH is the more reliable and sensitive assay.

*2) For Figure 3 – what is the affinity of NTE (1-42 aa's), for different nucleotide substrates? It will be good control to understand if NTE, by itself, has any nucleotide binding capacity.*

We agree this in an important consideration, and have purified MBP-SUV39H1 1-41 and measured its binding to the 19mer ssRNA (updated Figure 3). We found that 1-41 exhibits greatly reduced RNA binding affinity (51 μM) compared to either of the truncations, including the chromodomain (1-106: 0.15 μM, 42-106: 2.3 μM). However, it is interesting that the NTE is making an additional contribution to nucleic acid binding, and we plan to explore this effect further in the future.

*3)Subsection “RNA associates with the pericentric regions of human mitotic chromosomes”, last paragraph: Are there other genomic regions where the RNA is associated with mitotic chromosomes in cis? If yes, then what are those euchromatic/heterochromatic regions?*

We are also very interested in the RNAs bound to mitotic chromosomes, and we hope to further explore the identities and functions of these RNA in the future. We have begun preliminary experiments purifying mitotic chromosomes from human cells in the hopes of analyzing bound RNAs by RNA-seq. In addition, our lab has developed a new technique, ChAR-seq, to map chromatin-associated RNAs genome-wide (Bell et al., 2017) (http://biorxiv.org/content/early/2017/03/20/118786), and we plan to apply this technique to mitotic cell populations. However, we feel that a comprehensive characterization of other genomic regions that might have cis-associated noncoding RNAs is outside the scope of this manuscript. We plan to explore this in a future study using the new methods we are currently developing.

*4) Figure 6 – is the methylation efficiency of R55A and R84A mutants comparable to that of wild type protein? This will also impact results presented in Figure 6—figure supplement 1.*

We have included a new supplemental figure comparing the ability of SUV39H1 mutants to methylate histone H3/H4 tetramer in vitro (Figure 6—figure supplement 1). The data show comparable methyltransferase activity between WT and mutants; however, our purified full-length SUV39H1 contains degradation products that make us wary of making quantitative comparisons. As observed in previous studies with other histone methyltransferases, we see that the addition of RNA to our reaction inhibits SUV39H1 methyltransferase activity (Cifuentes-Rojas et al., 2014; Kaneko et al., 2014). However, we see that this inhibition occurs with our nucleic acid binding mutants as well, suggesting that this RNA-dependent inhibition of methyltransferase activity is largely non-specific.

*5) Subsection “A model for the RNA-dependent stabilization of SUV39H1 on chromatin”, first paragraph. The discussion of the pombe HP1/Swi6 and its requirement for Clr4-dependent H3K9me is incorrect. Even though this was shown in the referenced paper, subsequent work using the complete deletion of swi6 (instead of a this special swi6 mutant) have shown that loss of Swi6 has little impact on Clr4-dependent H3K9me (Sadaie et al. 2004 EMBO; Motamedi et al. 2008 Mol Cell).*

We thank the reviewers for alerting us to this important point. We have now corrected this sentence to more accurately describe the data in the literature.

6) Figure 6—figure supplement 1. Please show error bars for wt and calculate p values for these data.

We have now included error bars for the SUV39H1 WT measurement and calculate p values to describe HP1α localization defects in SUV39H1 mutants compared to WT.

*7) Subsection “SUV39H1 depends on direct RNA binding for its stable association with heterochromatin”, first paragraph. Please keep mobile/immobile percentages consistent between wt and mutant proteins. For example, convert 42% immobile fraction to 58% mobile considering the mobile fraction of mutants is indicated.*

We have now changed all FRAP descriptions in the text to consistently refer to the mobile fraction.